# Convergence of Mean-Field Langevin Stochastic Gradient Descent-Ascent for Distributional Minimax Optimization

Zhangyi Liu [1]  Feng Liu [2]  Rui Gao [3]  Shuang Li [4]

## Abstract

We study convergence properties of the discrete-time Mean-Field Langevin Stochastic Descent-Ascent (MFL-SDA) algorithm for solving distributional minimax optimization. These problems arise in various applications, such as zero-sum games, generative adversarial networks and distributionally robust learning. Despite the significance of MFL-SDA in these contexts, the discrete-time convergence rate remains underexplored. To address this gap, we establish a last-iterate convergence rate of $O(\frac{1}{\epsilon} \log \frac{1}{\epsilon})$ for MFL-SDA. This rate is nearly optimal when compared to the complexity lower bound of its Euclidean counterpart. This rate also matches the complexity of mean-field Langevin stochastic gradient descent for distributional minimization and the outer-loop iteration complexity of an existing double-loop algorithm for distributional minimax problems. By leveraging an elementary analysis framework that avoids PDE-based techniques, we overcome previous limitations and achieve a faster convergence rate.

## 1. Introduction

In this paper, we study a distributional minimax optimization of the form

$$\min_{\mu \in \mathcal{P}_1} \max_{\nu \in \mathcal{P}_2} E(\mu, \nu), \qquad (1)$$

where $\mathcal{P}_1, \mathcal{P}_2$ are sets of probability measures and $E$ is a convex-concave probability functional. This formulation generalizes minimax optimization problems in Euclidean spaces and arises widely in many important applications, including generative adversarial networks (GANs) (Goodfellow et al., 2020; Arjovsky et al., 2017), distributionally robust learning (Mądry et al., 2017), as well as zero-sum games with mixed Nash equillibrium (Daskalakis & Panageas, 2018).

Solving such optimization problems over distributional spaces is inherently challenging. Unlike in Euclidean spaces, the convexity/concavity of the objective function does not directly guarantee the convergence of gradient-based methods. To address this, mean-field Langevin dynamics (MFLD) has emerged as a powerful theoretical framework (Mei et al., 2018; Sirignano & Spiliopoulos, 2020; Hu et al., 2021; Chizat, 2022; Nitanda et al., 2022a; Suzuki et al., 2024). In this framework, an entropy-regularized distributional convex minimization is considered:

$$\min_{\mu \in \mathcal{P}(\Theta)} E(\mu) - \tau \mathcal{H}(\mu),$$

where $\mathcal{H}(\mu) = -\mathbb{E}_\mu[\log \mu]$ is the entropy functional. The mean-field Langevin gradient descent method in discrete time proceeds as follows:

$$\theta_{k+1} = \theta_k - \eta \nabla \frac{\delta E}{\delta \mu}[\mu_k](\theta_k) + \sqrt{2\eta\tau} \xi_k,$$

where $\mu_k$ is the distribution of $\theta_k$; $\frac{\delta E}{\delta \mu}[\mu_k]$ is the first variation of $E$ with respect to $\mu$ at $\mu_k$; and $\xi_k$ is an independent injected standard Gaussian noise. Notably, the gradient in this context is often replaced with an unbiased stochastic gradient estimator to accommodate practical settings such as stochastic gradient descent. By leveraging the uniform log-Sobolev inequality, corresponding to the Polyak-Łojasiewicz (PL) condition (Karimi et al., 2016) in the distributional space, recent works have established exponential (linear) convergence for MFLD in continuous time (Chizat, 2022; Nitanda et al., 2022a) sublinear convergence of the order $O(\frac{1}{\epsilon} \log \frac{1}{\epsilon})$ in discrete time (Nitanda et al., 2022a) and with a stochastic gradient oracle (Suzuki et al., 2024).

The MFLD framework can be extended to distributional minimax problems (1), giving rise to the Mean-Field Langevin Descent-Ascent (MFL-DA) algorithm. For instance, under

[1]Department of Mathematical Sciences, Tsinghua University, Beijing, China [2] School of Economics and Management, University of the Chinese Academy of Sciences, Beijing, China [3]Department of Information, Risk, and Operations Management, University of Texas at Austin, Austin, USA [4]School of Data Science, The Chinese University of Hong Kong (Shenzhen), Shenzhen, China. Correspondence to: Rui Gao <rui.gao@mccombs.utexas.edu>.

*Proceedings of the 42$^{nd}$ International Conference on Machine Learning*, Vancouver, Canada. PMLR 267, 2025. Copyright 2025 by the author(s).

two-sided Polyak-Łojasiewicz conditions, Lu (2023) established a linear convergence rate for MFL-DA in continuous time. Furthermore, Kim et al. (2024) proposed two variants: the Mean-Field Langevin Averaged Gradient method, which guarantees average-iterate convergence, and the Mean-Field Langevin Anchored Best Response, a symmetric double-loop algorithm whose outer loop achieves linear last-iterate convergence. Despite recent advances, the convergence properties of single-loop discrete-time algorithms remain largely unexplored. A notable exception is the discrete-time stochastic Mean-Field Langevin Averaged Gradient algorithm introduced by Kim et al. (2024), which achieves an average-iterate convergence rate of $O(\epsilon^{-O(1/\alpha)})$, where $\alpha$ is the log-Sobolev constant. However, this rate appears suboptimal. Furthermore, last-iterate convergence is often more desirable in minimax optimization due to its practical significance. To our knowledge, the nearly optimal complexity of last-iterate convergence for discrete-time Mean-Field Langevin Stochastic Descent-Ascent (MFL-SDA) has not yet been investigated.

In contrast, minimax problems in Euclidean spaces have been extensively studied. For stochastic strongly convex-strongly concave functions, an optimal complexity of $O(1/\epsilon)$ is achieved by a variant of stochastic gradient descent-ascent (Yan et al., 2020; Zhang & Hu, 2025). Similarly, for problems with two-sided PL conditions using a stochastic oracle, an $O(1/\epsilon)$ convergence rate is attained by alternating stochastic gradient descent-ascent (Yang et al., 2020). In view of this, it is natural to ask whether MFL-SDA can achieve a similar last-iterate convergence rate in the distributional space under the two-sided distributional PL condition. These gaps in understanding motivate our central question:

*Can the discrete time MFL-SDA for (1) achieve a last-iterate complexity similar to that of minimax problems in the Euclidean space?*

## 1.1. Related Work

**Mean Field Langevin Dynamics**   Our primary motivation stems from recent advancements in applying MFLD to neural networks. The pioneering works of Mei et al. (2018); Chizat & Bach (2018); Sirignano & Spiliopoulos (2020) leveraged MFLD to establish global convergence guarantees for (noisy) gradient descent in optimizing two-layer neural networks. Building on these foundations, Chizat (2022); Nitanda et al. (2022a) demonstrated that, under the log-Sobolev inequality–the distributional counterpart of the PL condition, continuous-time MFLD for distributional minimization can achieve an exponential convergence rate. Furthermore, Nitanda et al. (2022a) showed that the discrete-time MFLD enjoys a convergence guarantee with a complexity of $O(\frac{1}{\epsilon} \log(\frac{1}{\epsilon}))$. This rate also extends to set-

tings involving stochastic gradient oracles and finite-particle approximations, as established by Suzuki et al. (2024). Our discrete-time analysis draws inspiration from a recent work by Wang et al. (2024), which established the convergence of MFLD in discrete time for one-hidden layer neural network with softmax activations and a square loss.

**Distributional Minimax Optimization**   Compared to distributional minimization problems, the study of MFLD in distributional minimax problems remains limited, particularly in the discrete-time setting. The most relevant work in this area is by Kim et al. (2024), who investigated the convergence of general convex-concave functionals in both continuous and discrete time. Rather than analyzing the standard MFL-DA algorithm, they proposed two alternative methods: the Mean-Field Langevin Averaged Gradient (MFL-AG) (Tao et al., 2021) and the Mean-Field Langevin Anchored Best Response (MFL-ABR) algorithm (Lascu et al., 2023). MFL-AG updates are computed using a weighted average of past gradients instead of the current gradient. For the discrete-time setting with a stochastic oracle, they demonstrated that the average-iterate convergence rate to an $\epsilon$-optimal saddle point is $O(\epsilon^{-1-O(1/\alpha)})$, where $\alpha$ is the log-Sobolev constant. They derived an $O(1/d)$ bound for $\alpha$ and noted that this dependence on the dimension of the sample space can sometimes be avoided. MFL-ABR is a double-loop algorithm, with the outer loop having an iteration complexity of $O(\frac{1}{\epsilon} \log \frac{1}{\epsilon})$. Additionally, Cai et al. (2024) analyzed MFL-SDA with a bilinear functional and a strongly convex-concave interaction function, an assumption not applicable in our context. Wang & Chizat (2022) introduced and analyzed a particle-based method inspired by the mirror prox algorithm in Euclidean space.

In addition, there has been some convergence analysis for MFL-DA in continuous time. Domingo-Enrich et al. (2020), Ma & Ying (2021) and Lu (2023) studied its convergence for finding the saddle point of an entropy-regularized objective. Qualitative convergence results were established in Domingo-Enrich et al. (2020), while Ma & Ying (2021) proved the asymptotic convergence under quasi-static conditions, where the ascent dynamics is infinitely faster or slower than the descent dynamics. Lu (2023) provided non-asymptotic exponential convergence rates with based on a two-sided PL condition. Zhu et al. (2024) demonstrated a sublinear convergence rate for the stochastic gradient descent-ascent algorithm for solving functional minimax optimization using mean-field neural networks, and showed that the discrete-time algorithm converges to its continuous-time counterpart.

**Eucliean Minimax Optimization**   There is a substantial body of research on minimax optimization problems in Euclidean space (e.g., see recent works of Daskalakis &

Panageas (2018); Doan (2022); Jin et al. (2020); Lin et al. (2020); Li et al. (2022)), encompassing both the convex-concave setting and more general scenarios. Among these studies, some provide valuable insights for our work. For instance, Doan (2022) analyzed the convergence properties in Euclidean space under the PL condition using continuous-time analysis. Similarly, Yang et al. (2020) proposed a (stochastic) alternating gradient descent-ascent algorithm under the PL condition. We aim to achieve a similar convergence result in the distributional space.

## 1.2. Main contributions

We establish the last-iterate convergence rate of the MFL-SDA algorithm in discrete time. Our $O(\frac{1}{\epsilon} \log \frac{1}{\epsilon})$ bound is nearly optimal, comparable to the $O(\frac{1}{\epsilon})$ bound for minimax problems in Euclidean space, and aligns with the $O(\frac{1}{\epsilon} \log \frac{1}{\epsilon})$ complexity of mean-field Langevin stochastic gradient descent in distributional minimization problems (Suzuki et al., 2024) and the outer-loop iteration complexity of the double-loop MFL-ABR algorithm (Kim et al., 2023).

Unlike previous studies that rely on PDE-based techniques, our proof is elementary and bears more resemblance to analyses conducted in Euclidean space. Our perturbation analysis is flexible enough to overcome the limitations of existing discrete-time algorithm analyses and thereby improves the convergence rate.

We apply our findings to several applications, including zero-sum games and mean field neural networks, and verify the essential assumptions required to ensure the theoretical convergence of MFL-SDA in these contexts.

## 2. Preliminaries

### 2.1. Problem Setup

Let $\mathcal{P}(\mathbb{R}^{d_\Theta})$ (resp., $\mathcal{P}(\mathbb{R}^{d_\Omega})$) denote the space of probability distributions on $\mathbb{R}^{d_\Theta}$ (resp., $\mathbb{R}^{d_\Omega}$) where the entropy and second-order moment are well-defined. Let $J : \mathcal{P}(\mathbb{R}^{d_\Theta}) \times \mathcal{P}(\mathbb{R}^{d_\Omega}) \to \mathbb{R}$ be a convex-concave functional, in the sense that for any distributions $\mu_1, \mu_2 \in \mathcal{P}(\mathbb{R}^{d_\Theta})$ and $\nu_1, \nu_2 \in \mathcal{P}(\mathbb{R}^{d_\Omega})$ and any $t \in [0, 1]$, the following conditions hold

$$J(t\mu_1 + (1-t)\mu_2, \nu_1) \le tJ(\mu_1, \nu_1) + (1-t)J(\mu_2, \nu_1),$$
$$J(\mu_1, t\nu_1 + (1-t)\nu_2) \ge tJ(\mu_1, \nu_1) + (1-t)J(\mu_1, \nu_2).$$

We consider the following energy functional

$$E(\mu, \nu) := J(\mu, \nu) + \tau \mathsf{KL}(\mu|\rho^\mu) - \tau \mathsf{KL}(\nu|\rho^\nu), \quad (2)$$

where $\mathsf{KL}(\mu|\rho^\mu) = \mathbb{E}_\mu \left[ \log \left( \frac{d\mu}{d\rho^\mu} \right) \right]$ denotes the Kullback-Leibler (KL) divergence; the reference distributions $\rho^\mu, \rho^\nu$ are assumed to be standard Gaussian for simplicity, though they can be generalized to distributions with a strongly convex potential. The hyperparameter $\tau > 0$ controls the regularization. This setup aligns with Kim et al. (2023) but

differs slightly from Lu (2023), who considered entropy regularization. When the reference distributions are standard Gaussians, the KL divergence is equivalent to the entropy plus the second-order moment of the distribution. With KL (or entropy) regularization, the energy functional $E(\mu, \nu)$ is strongly convex in $\mu$ and strongly concave in $\nu$, which ensures the existence and uniqueness of the mixed Nash equilibrium $(\mu^*, \nu^*)$ (Kim et al., 2024, Proposition 2.1), where

$$E(\mu^*, \nu) \le E(\mu^*, \nu^*) \le E(\mu, \nu^*), \quad \forall \mu, \nu. \quad (3)$$

Denote by $\frac{\delta J}{\delta \mu}[\mu, \nu](\cdot)$ and $\frac{\delta J}{\delta \nu}[\mu, \nu](\cdot)$ the first variations of $J$ with respect to $\mu$ and $\nu$, respectively, which are assumed to be well-defined throughout. Convexity/concavity can be also defined via first variations, as detailed in Appendix A.1. Denote by $\theta$ and $\omega$ the random variables with distributions $\mu$ and $\nu$, respectively. Denote by $\nabla \frac{\delta J}{\delta \mu}[\mu, \nu](\cdot)$ and $\nabla \frac{\delta J}{\delta \nu}[\mu, \nu](\cdot)$ the Wasserstein gradients of $J$ with respect to $\mu$ and $\nu$, respectively. For more details on Wasserstein gradient flow, see Ambrosio et al. (2008); Santambrogio (2015) .

The continuous-time mean-field gradient flow of (2) is given by

$$d\theta_t = -\nabla \frac{\delta J}{\delta \mu}[\mu_t, \nu_t](\theta_t)dt - \tau\theta_t dt + \sqrt{2\tau}dB_t^1,$$

$$d\omega_t = \eta \cdot \left( \nabla \frac{\delta J}{\delta \nu}[\mu_t, \nu_t](\omega_t)dt - \tau\omega_t dt + \sqrt{2\tau}dB_t^2 \right). \tag{4}$$

Here, $\mu_t$ and $\nu_t$ are the distributions of $\theta_t$ and $\omega_t$ at time $t$, respectively. The drift terms $\nabla \frac{\delta J}{\delta \mu}[\mu_t, \nu_t](\theta_t)$ and $\nabla \frac{\delta J}{\delta \nu}[\mu_t, \nu_t](\omega_t)$ are the Wasserstein gradients with respect to $\mu_t$ and $\nu_t$ at time $t$, while $\{B_t^1\}_t, \{B_t^2\}_t$ are two independent Brownian motions initialized at zero. The decay terms $\tau\theta_t$ and $\tau\omega_t$ correspond to the second-order moment regularization associated with the KL regularization. To simplify the presentation, we set the scaling factor and the weight decay to be the same as the hyperparameter for KL regularization, corresponding to the standard Gaussian, but our analysis can be easily extended to other choices. The scaling factor $\eta > 0$ follows from the formulation in Lu (2023), who derived an exponential convergence of (4) to the saddle point.

---

**Algorithm 1** Mean field Langevin Stochastic Descent-Ascent (MFL-SDA)

---
1: Initialize $\mu_0, \nu_0, K$
2: **for** $k = 1$ to $K - 1$ **do**
3:     $\theta_{k+1} \leftarrow \theta_k - \eta_1(\hat{\nabla}\frac{\delta J}{\delta \mu}[\mu_k, \nu_k](\theta_k) + \tau\theta_k) + \sqrt{2\eta_1\tau}\xi_k^1$
4:     $\omega_{k+1} \leftarrow \omega_k + \eta_2(\hat{\nabla}\frac{\delta J}{\delta \nu}[\mu_{k+1}, \nu_k](\omega_k) - \tau\omega_k) + \sqrt{2\eta_2\tau}\xi_k^2$
5: **end for**

---

In this paper, we focus on the discrete-time counterpart of (4), as outlined in Algorithm 1. In this algorithm, $\hat{\nabla}$ denotes

an unbiased stochastic gradient estimator, and $\xi_k^1$ and $\xi_k^2$ are independent standard normal random variables sampled at each iteration $k$. The algorithm employs dynamics with two timescales, $\eta_1, \eta_2 > 0$, aligning with the continuous-time updates in Lu (2023) and the discrete-time Euclidean updates in Yang et al. (2020). Additionally, the updates for $\mu$ and $\nu$ are performed alternately, following a strategy similar to Yang et al. (2020) for Euclidean spaces. Our primary goal of this work is to analyze the convergence of Algorithm 1 to the saddle point $(\mu^*, \nu^*)$.

## 2.2. Log-Sobolev Inequality

Unlike minimax problems in Euclidean space, the convex-concave property of the energy functional does not provide immediate algorithmic benefits for gradient-based methods in the Wasserstein space[1]. Conversely, a more generalized notion, the Polyak-Łojasiewicz (PL) condition, has proven useful for analyzing the gradient flow in the distribution space. In Euclidean space, the PL condition facilitates a linear convergence rate (Karimi et al., 2016). Extending this notion to distribution spaces leads to the log-Sobolev inequality, which has been employed in the analysis of optimization over distribution spaces with MFLD (Chizat, 2022; Nitanda et al., 2022a; Lu, 2023; Kim et al., 2023).

**Definition 1.** A distribution $\nu$ satisfies the log-Sobolev inequality with parameter $\alpha > 0$ if, for all $\mu \ll \nu$, the following holds:

$$\mathsf{KL}(\mu|\nu) \le \frac{1}{2\alpha} I(\mu|\nu), \tag{LSI}$$

where the relative Fisher information $I(\mu|\nu)$ is defined as

$$I(\mu|\nu) := \mathbb{E}_\mu\left[\left\|\nabla \log \frac{d\mu}{d\nu}\right\|^2\right].$$

## 2.3. Gibbs Distributions

To define the optimizer of the inner maximization in (2), we introduce several Gibbs distributions. We define the Gibbs operators $\mathcal{K}_\mu^+[\cdot]$ and $\mathcal{K}_\nu^-[\cdot]$ as

$$\mathcal{K}_\mu^+[\nu](\omega) \propto \exp\left(\tau^{-1}\frac{\delta J}{\delta \nu}[\mu,\nu](\omega) - \frac{\|\omega\|_2^2}{2}\right), \tag{5}$$

$$\mathcal{K}_\nu^-[\mu](\theta) \propto \exp\left(-\tau^{-1}\frac{\delta J}{\delta \mu}[\mu,\nu](\theta) - \frac{\|\theta\|_2^2}{2}\right). \tag{6}$$

When the energy functional $J(\mu,\nu)$ is bilinear in $\mu$ and $\nu$, Lu (2023) demonstrated that $\mathcal{K}_\mu^+[\nu] = \arg\max_{\nu \in \mathcal{P}(\mathbb{R}^{d_\Omega})} E(\mu,\nu)$. However, this equality no longer holds when $J(\mu,\nu)$ is a general nonlinear functional.

[1]In fact, another notion of convexity, known as displacement convexity or geodesic convexity, is more applicable for studying gradient flow in the Wasserstein space (Villani, 1998; Ambrosio et al., 2008), which is not satisfied in our setting.

To address this, we consider the fixed point of (5), defined as

$$\mathcal{K}_*^+[\mu](\omega) := \frac{1}{\mathcal{Z}^*(\mu)} \exp\left(\tau^{-1}\frac{\delta J}{\delta \nu}[\mu, \mathcal{K}_*^+[\mu]](\omega) - \frac{\|\omega\|_2^2}{2}\right),$$

where

$$\mathcal{Z}^*(\mu) := \int_{\mathbb{R}^{d_\Omega}} \exp\left(\tau^{-1}\frac{\delta J}{\delta \nu}[\mu, \mathcal{K}_*^+[\mu]](\omega) - \frac{\|\omega\|_2^2}{2}\right)d\omega$$

is a normalization constant. It can be verified that $\mathcal{K}_*^+[\mu]$ satisfies the following equation of $\nu$:

$$\frac{\delta J}{\delta \nu}[\mu,\nu](\omega) - \tau\frac{\|\omega\|_2^2}{2} - \tau\log\nu(\omega) = \text{const}, \ \forall\omega,$$

where the constant is independent of $\omega$ but may depend on $\mu$. Observe that the above equation is the first-order condition for the problem $\max_{\nu \in \mathcal{P}(\mathbb{R}^{d_\Omega})} E(\mu,\nu)$. Thus we have

$$\mathbb{E}[\mu, \mathcal{K}_*^+[\mu]] = \max_{\nu \in \mathcal{P}(\mathbb{R}^{d_\Omega})} E(\mu,\nu) =: E^*(\mu). \tag{7}$$

This will be frequently used in our convergence analysis.

# 3. Main Results

The goal of this section is to establish the convergence of MFL-SDA (Algorithm 1). Following Yang et al. (2020); Lu (2023), we introduce the Lyapunov function

$$\mathcal{L}(\mu,\nu) := \mathcal{L}_1(\mu) + \lambda\mathcal{L}_2(\mu,\nu),$$

where $\lambda > 0$ is a fixed constant, and

$$\mathcal{L}_1(\mu) := \max_{\nu' \in \mathcal{P}(\mathbb{R}^{d_\Omega})} E(\mu,\nu') - \min_{\mu' \in \mathcal{P}(\mathbb{R}^{d_\Theta})} \max_{\nu' \in \mathcal{P}(\mathbb{R}^{d_\Omega})} E(\mu',\nu'),$$

$$\mathcal{L}_2(\mu,\nu) := \max_{\nu' \in \mathcal{P}(\mathbb{R}^{d_\Omega})} E(\mu,\nu') - E(\mu,\nu). \tag{8}$$

Note that $\mathcal{L}_1$ and $\mathcal{L}_2$ are both non-negative and vanish if and only if $(\mu,\nu) = (\mu^*,\nu^*)$ in the weak sense.

We begin by stating our main assumptions.

## 3.1. Assumptions

We impose the following assumptions.

**Assumption 1** (Initial condition). The initial iterate $(\mu_0, \nu_0)$ satisfies $E(\mu_0, \nu_0) < \infty$, and the initial third-order moments satisfy $\mathbb{E}_{\mu_0}[\|\theta_0\|_2^4], \mathbb{E}_{\nu_0}[\|\omega_0\|_2^4] < \infty$.

**Assumption 2** (Regularity of functional $J$). The functional $J(\mu,\nu)$ is convex in $\mu$ and concave in $\nu$. The first variations of $J$ have bounded derivatives up to the fourth order: $\|\nabla^i\frac{\delta J}{\delta \mu}\|_F, \|\nabla^i\frac{\delta J}{\delta \nu}\|_F \le M_i$, $i = 1,\ldots,4$, where $\|\cdot\|_F$ is Frobenius norm. Additionally, $J$ has a bounded cross second-order variation: $\|\frac{\delta^2 J}{\delta\mu\delta\nu}\|_\infty \le C_0$. Moreover, the Hessian of its second variations are bounded: $\|\nabla_\theta\nabla_{\theta'}^\top\frac{\delta^2 J}{\delta\mu^2}\|_F, \|\nabla_\theta\nabla_\omega^\top\frac{\delta^2 J}{\delta\mu\delta\nu}\|_F, \|\nabla_\omega\nabla_{\omega'}^\top\frac{\delta^2 J}{\delta\nu^2}\|_F \le C_1$, $\|\nabla_\theta\frac{\delta^2 J}{\delta\mu\delta\nu}\|_\infty, \|\nabla_\omega\frac{\delta^2 J}{\delta\mu\delta\nu}\|_\infty \le C_2$.

**Assumption 3** (Log-Sobolev inequality). For any $\mu \in \mathcal{P}(\mathbb{R}^{d_\Theta}), \nu \in \mathcal{P}(\mathbb{R}^{d_\Omega})$, the measures $\mathcal{K}_\mu^+[\nu], \mathcal{K}_\nu^-[\mu]$ satisfy LSI with parameter $\alpha$.

We will validate these assumptions in various applications, as discussed in Section 4 and Appendix B.1.

*Remark* 1. Using Suzuki et al. (2024); Kim et al. (2023), Asssumption 2 implies Assumption 3 with a conservative LSI constant; see Lemma 5 in the Appendix.

*Remark* 2. For simplicity, we assume $d_\Theta = d_\Omega = d$ in proofs related to convergence rates.

### 3.2. Convergence Analysis

At a high level, our analysis framework parallels that of stochastic gradient descent-ascent in Euclidean spaces (e.g., Yang et al. (2020)). However, analyzing the evolution of distributions in our problem presents greater challenges. Specifically, the noise terms $\xi_k^1$ and $\xi_k^2$ introduce time-discretization error terms that do not appear in Euclidean problems, even when the gradient oracle is exact. To show that these errors are negligible higher-order terms, most existing analyses leverage the connection between discrete-time updates and continuous-time gradient flow (4) (e.g., Vempala & Wibisono (2022); Nitanda et al. (2022a); Suzuki et al. (2024); Kim et al. (2024)). In contrast, we directly analyze the discrete-time updates and carefully bound the higher-order terms in Taylor expansions through integration by parts and the divergence theorem. As will be seen from Theorem 2, our approach is flexible enough to handle the stochastic gradient oracle in a straightforward manner, similar to the Euclidean case.

Below, to illustrate the main idea, we first present the convergence analysis of MFL-DA, where the algorithm has access to exact gradients, then extend our results to MFL-SDA in Section 3.3.

Observe that

$$\mathcal{L}(\mu_{k+1}, \nu_{k+1}) - \mathcal{L}(\mu_k, \nu_k)$$
$$= \mathcal{L}_1(\mu_{k+1}) - \mathcal{L}_1(\mu_k) + \lambda(\mathcal{L}_2(\mu_{k+1}, \nu_{k+1}) - \mathcal{L}_2(\mu_k, \nu_k)).$$

In the following, we bound the two terms $\mathcal{L}_1(\mu_{k+1}) - \mathcal{L}_1(\mu_k)$ and $\mathcal{L}_2(\mu_{k+1}, \nu_{k+1}) - \mathcal{L}_2(\mu_k, \nu_k)$ in Sections 3.2.1 and 3.2.2, respectively. Once these bounds are established, we sum them over all iterations and apply a telescoping argument to derive the overall convergence rate in Section 3.2.3.

To simplify the presentation, we define

$$g_k := \nabla \frac{\delta J}{\delta \mu}[\mu_k, \nu_k](\theta_k) + \tau\theta_k,$$

$$h_k := \nabla \frac{\delta J}{\delta \nu}[\mu_{k+1}, \nu_k](\omega_k) - \tau\omega_k,$$

$$f_k := \nabla \frac{\delta J}{\delta \mu}[\mu_k, \mathcal{K}_*^+[\mu_k]](\theta_k) + \tau\theta_k,$$

where $f_k$, depending on the Gibbs distribution $\mathcal{K}_*^+[\mu_k]$, is only used for theoretical analysis but not in the implementation of algorithm. Moreover, we omit the constants appearing in the higher-order error bounds, whose explicit expressions are provided in Appendix A.3.

#### 3.2.1. BOUNDING $\mathcal{L}_1(\mu_{k+1}) - \mathcal{L}_1(\mu_k)$

By definition of $\mathcal{L}_1$ in (8) and $E^*$ in (7), it holds that

$$\mathcal{L}_1(\mu_{k+1}) - \mathcal{L}_1(\mu_k) = E^*(\mu_{k+1}) - E^*(\mu_k).$$

We have the following result.

**Proposition 1.** *Assume Assumptions 1-3 hold. Let $\eta_1 < \frac{1}{C_1}$. Then it holds that*

$$\mathcal{L}_1(\mu_{k+1}) - \mathcal{L}_1(\mu_k) \leq$$
$$-\frac{\eta_1}{2}\left(\mathbb{E}_{\mu_k}[\|f_k + \nabla\log\mu_k\|_2^2] - \mathbb{E}_{\mu_k}[\|g_k - f_k\|_2^2]\right) + \Gamma_0\eta_1^2,$$

*where remainder $\Gamma_0$ is defined in Appendix A.3.*

Proposition 1 establishes that the difference $\mathcal{L}_1(\mu_{k+1}) - \mathcal{L}_1(\mu_k)$, or equivalently, $E^*(\mu_{k+1}) - E^*(\mu_k)$, is bounded by the squared Wasserstein gradient norm of $E^*(\mu)$ at $\mu_k$. The bound consists of two dominant terms, corresponding to the squared norm of the partial Wasserstein gradients of $E(\mu, \mathcal{K}_*^+[\mu])$—an equivalent form of $E^*(\mu)$—with respect to the first and second arguments of the energy functional $E$. The first term arises from the outer minimization problem in (2) with respect to $\mu$, while the second term serves as a correction due to the inner maximization in (2) over $\nu$. This result parallels the convergence analysis in Euclidean space (Yang et al., 2020). However, obtaining the $O(\eta_1^2)$ remainder requires significant effort, as it involves analyzing the smoothness of the functional $E^*(\mu)$, which, in turn, depends on the smoothness of the operator $\mathcal{K}_*^+[\mu]$; see Lemma 9 in the Appendix for details.

#### 3.2.2. BOUNDING $\mathcal{L}_2(\mu_{k+1}, \nu_{k+1}) - \mathcal{L}_2(\mu_k, \nu_k)$

By definition, we have

$$\mathcal{L}_2(\mu_{k+1}, \nu_k) - \mathcal{L}_2(\mu_k, \nu_k)$$
$$= \left(E^*(\mu_{k+1}) - E^*(\mu_k)\right) + \left(E(\mu_k, \nu_k) - E(\mu_{k+1}, \nu_k)\right).$$

We have already established an upper bound on the first difference above in Proposition 1. It remains to bound the second difference.

**Lemma 1.** *Assume Assumptions 1-3 hold. Then we have*

$$E(\mu_{k+1}, \nu_{k+1}) - E(\mu_{k+1}, \nu_k)$$
$$\geq \eta_2 \mathbb{E}_{\nu_k}[\|h_k - \tau\nabla\log\nu_k\|_2^2] - \Gamma_2\eta_2^2.$$

*Similarly, we have*

$$E(\mu_{k+1}, \nu_k) - E(\mu_k, \nu_k)$$
$$\leq -\eta_1 \mathbb{E}_{\mu_k}[\|g_k + \nabla\log\mu_k\|_2^2] + \Gamma_1\eta_1^2.$$

*Here, the remainders $\Gamma_1$ and $\Gamma_2$ are defined in Appendix A.3.*

This lemma provides a lower bound on the per-step objective improvement for the inner gradient ascent and outer gradient descent in solving (2). The leading terms in both bounds correspond to the squared Wasserstein gradient norm with respect to $\nu_k$ and $\mu_k$, respectively. The $O(\eta_1^2)$ and $O(\eta_2^2)$ bias terms arise from time discretization and are consistent with the results of Nitanda et al. (2022a) for distributional convex minimization problems.

Using Lemma 1 and Assumption 3, the following result is immediate.

**Lemma 2.** *Assume Assumptions 1-3 hold. Then we have*

$$\mathcal{L}_2(\mu_{k+1}, \nu_{k+1}) \leq (1 - 2\eta_2 \tau \alpha)\mathcal{L}_2(\mu_{k+1}, \nu_k) + \Gamma_2 \eta_2^2.$$

Using the bound in Proposition 1 and combining Lemma 1 and Lemma 2, we obtain the following bound on the difference between $\mathcal{L}_2(\mu_{k+1}, \nu_{k+1})$ and $\mathcal{L}_2(\mu_k, \nu_k)$.

**Proposition 2.** *Assume Assumptions 1-3 hold. Then we have*

$$\mathcal{L}_2(\mu_{k+1}, \nu_{k+1}) \leq$$
$$(1 - 2\eta_2 \tau \alpha)\Big(\mathcal{L}_2(\mu_k, \nu_k) + \eta_1 \mathbb{E}_{\mu_k}[\|g_k + \nabla \log \mu_k\|_2^2]$$
$$- \frac{\eta_1}{2}\mathbb{E}_{\mu_k}[\|f_k + \nabla \log \mu_k\|_2^2] + \frac{\eta_1}{2}\mathbb{E}_{\mu_k}[\|g_k - f_k\|_2^2]\Big)$$
$$+ \Gamma_2 \eta_2^2 + (1 - 2\eta_2 \tau \alpha)(\Gamma_1 + \Gamma_0)\eta_1^2.$$

Combining Propositions 1 and 2 yields a recursive bound on the Lyapunov function $\mathcal{L}$.

### 3.2.3. CONVERGENCE OF MFL-DA

By applying Propositions 1 and 2, and using an argument similar to that in the Euclidean case (Yang et al., 2020), we can establish the following convergence result for MFL-DA. Note that different learning rates $\eta_1, \eta_2$ are employed to ensure convergence, as is commonly done (Yang et al., 2020; Lu, 2023).

**Theorem 1.** *Assume Assumptions 1-3 hold. Set $\tau < \frac{1}{2C_1^2}$, $\eta_1 \leq \frac{1}{C_1}, \eta_2 \leq \frac{1}{2\tau\alpha}$ and $\eta_1 = \min\{\lambda, 0.2, \frac{1}{\tau\alpha}\}\tau\alpha\eta_2$, then it holds that*

$$\mathcal{L}(\mu_K, \nu_K) \leq (1 - 2\eta_1 \tau \alpha)^K \mathcal{L}(\mu_0, \nu_0) + R_1, \quad (9)$$

*where $R_1 = \frac{\lambda(\Gamma_2 \eta_2^2 + (1 - 2\eta_2 \tau \alpha)(\Gamma_1 + \Gamma_0)\eta_1^2) + \Gamma_1 \eta_1^2}{\eta_1 \tau \alpha}$.*

Theorem 1 demonstrates that the Lyapunov function $\mathcal{L}(\mu, \nu)$ converges to a bias $R_1$ at a geometric rate. This geometric decay aligns with the exponential decay observed in the continuous-time case, as established by Lu (2023). The bias term $R_1$, which results from time discretization, is of order $O(\eta_1)$. To assess the algorithm's complexity, since in practical algorithm we often assume $\tau, \eta_1, \eta_2$ is

small, then the remainder $r_{g4}, r_{h4}$ in Appendix A.3 caused by the fourth moment has $\max\{O(1), \tau^2 d^2\}$ scale. Substitute into $\Gamma_0, \Gamma_1, \Gamma_2$ we can get an estimation of these bias term: $\Gamma_0 = \max\{O(1), \tau d, \tau^2 d^2, \frac{d}{\alpha^{1/2}\tau}\}$, $\Gamma_{1(2)} = \max\{O(1), \tau d, \tau^2 d^2\}$. Replace them into $R_1$ we can get a worst bound $R_1 = O(\frac{d\eta_1}{\tau^3 \alpha^3})$. Let $R_1 = \epsilon$, then choose $\eta_1 = O(\frac{\epsilon \tau^3 \alpha^3}{d})$ to get a sample complexity $K = O(\frac{d}{\epsilon \tau^4 \alpha^4} \log \frac{1}{\epsilon})$. This complexity matches that of MFLD for distributional convex minimization (Nitanda et al., 2022a) and the outer-loop complexity of discrete-time MFL-ABR (Kim et al., 2024), and the sample complexity $K = O(\frac{d}{\epsilon \tau^2 \alpha^2} \log \frac{1}{\epsilon})$ in (Nitanda et al., 2022b) who discussed about discrete-time MFLD in the single minimization problem, the higher order of $\tau, \alpha$ is because the two-timescale optimization scheme in minimax problem, hence the efficiency of this algorithm mainly depends on the slower part, which is the descent part in our paper.

Comparatively, in the Euclidean case, the (exact) gradient descent-ascent method (Yang et al., 2020, Theorem 3.1 with $\sigma = 0$) achieves a linear convergence rate of $O(\frac{1}{\epsilon})$, where the bias term is absent. This is because, in the Euclidean setting, higher-order terms in the Taylor expansion can be absorbed into the first-order squared gradient norm, resulting in a contraction of the Lyapunov function. However, in our distributional case, the randomness introduced by Gaussian noise prevents the absorption of higher-order terms into the first-order term, leading to a sublinear convergence rate.

*Remark 3.* Similar to Lu (2023), we can also consider the Lyapunov function $\mathcal{L}_3(\nu) + \lambda\mathcal{L}_4(\mu, \nu)$, where

$$\mathcal{L}_3(\nu) := \max_{\nu' \in \mathcal{P}(\mathbb{R}^{d_\Omega})} \min_{\mu' \in \mathcal{P}(\mathbb{R}^{d_\Theta})} E(\mu', \nu') - \min_{\mu' \in \mathcal{P}(\mathbb{R}^{d_\Theta})} E(\mu', \nu),$$
$$\mathcal{L}_4(\mu, \nu) := E(\mu, \nu) - \min_{\mu' \in \mathcal{P}(\mathbb{R}^{d_\Theta})} E(\mu', \nu),$$

which is useful for max-min problem. The result is similar to Theorem 1 but with a reverse scaling of $\eta_1$ and $\eta_2$.

### 3.3. Convergence of MFL-SDA

In the previous subsection, we assumed the availability of exact gradients. In practice, however, we often work with stochastic gradients, where the exact gradient is replaced by an unbiased estimate. Thanks to the similar high-level structure of our analysis to the Euclidean case (Yang et al., 2020), our results can be extended to the stochastic gradient setting in a straightforward manner. We can show that the convergence rate of MFL-SDA is analogous to that of MFL-DA, with the same geometric decay rate and bias term. The main difference is that the bias term now depends on the higher-order moments of the stochastic gradients.

**Assumption 4** (Bounded moments). There exists $\zeta \geq 0$ such that

$$\mathbb{E}[\|\hat{g}_k - g_k\|^4 | \mu_k, \nu_k], \mathbb{E}[\|\hat{h}_k - h_k\|^4 | \mu_{k+1}, \nu_k] \leq \zeta,$$

We have the following result for MFL-SDA analogous to Theorem 1.

**Theorem 2.** *Under the same setup as in Theorem 1, and assume further that Assumption 4 holds. Then we have*

$$\mathbb{E}[\mathcal{L}(\mu_K, \nu_K)] \leq (1 - 2\eta_1\tau\alpha)^K \mathcal{L}(\mu_0, \nu_0) + R_2,$$

*where* $R_2 = \frac{\lambda(\hat{\Gamma}_2\eta_2^2 + (1-2\eta_2\tau\alpha)(\hat{\Gamma}_1+\hat{\Gamma}_0)\eta_1^2) + \hat{\Gamma}_1\eta_1^2}{\eta_1\tau\alpha}$ *with* $\hat{\Gamma}_0$, $\hat{\Gamma}_1$, $\hat{\Gamma}_2$ *defined in* (17)*, and the expectation is taken over the randomness in the stochastic gradients.*

This result demonstrates that the convergence rate of MFL-SDA (Algorithm 1) is $O(\frac{1}{\epsilon}\log\frac{1}{\epsilon})$. This rate is nearly optimal when compared to the $O(\frac{1}{\epsilon})$ convergence rate of stochastic gradient descent-ascent for minimax optimization with a two-sided PL condition in Euclidean space, as discussed in Yang et al. (2020).

*Proof Sketch.* Let $\mathscr{H}_k$ be $\sigma$-field generated by the random gradients up to iteration $k - 1$. Similar to the Euclidean case, at iteration $k$, we analyze the impact of using stochastic gradient oracles on the bound $\mathbb{E}[\mathcal{L}(\mu_{k+1}, \nu_{k+1}) \mid \mathscr{H}_k] - \mathcal{L}(\mu_k, \nu_k)$. By controlling the second- and third-order moments of the stochastic gradients, we ensure that the convergence properties established in Section 3.2.3 remain valid, albeit with additional error terms. Specifically, thanks to the unbiasedness of the stochastic gradients and the moment bounds in Assumption 4, the error term introduced by stochastic gradients is $O(\eta^2)$ or higher order. This holds for both the squared gradient norm and the entropy regularization. By carefully bounding these error terms, we show that the overall convergence result holds with a modified remainder term that accounts for the inexactness of the gradient. □

Furthermore, leveraging the connection between the Lyapunov function and KL-divergence, we can also obtain convergence rate in terms of KL-divergence to $(\mu^*, \nu^*)$ and further in terms of Wasserstein distance.

**Corollary 1** (Convergence in KL-divergence / Wasserstein distance). *Under the same setup as in Theorem 2, suppose that $\mathcal{K}_*^+[\mu]$ and $\mathcal{K}_*^-[\nu]$ satisfy LSI with constant $\alpha_1$, then it holds that*

$$\frac{2\tau}{\alpha_1}(\mathsf{W}_2^2(\mu_k, \mu^*) + \mathsf{W}_2^2(\nu_k, \nu^*))$$
$$\leq \tau(\mathsf{KL}(\mu_K|\mu^*) + \mathsf{KL}(\nu_K|\nu^*))$$
$$\leq (1 - \eta_1\tau\alpha)^K Q\mathcal{L}(\mu_0, \nu_0) + QR_1,$$

*where* $Q = \left(1 + (\frac{2}{\lambda} + \frac{4C_0^2}{\tau^2})\right)$ *and* $\mathsf{W}_2$ *denotes the 2-Wasserstein distance.*

Similar to Remark 1, Assumption 2 implies a conservative upper bound on the LSI constant $\alpha_1$. Kim et al. (2023)

established that the stochastic MLF-AG achieves a convergence rate of $O(\epsilon^{-1-O(\alpha^{-1})})$ in terms of the squared 1-Wasserstein distance, where $\alpha$ is the LSI constant in Assumption 3. Since $\mathsf{W}_2$ upper bounds $\mathsf{W}_1$, Corollary 1 improves the existing complexity bound for stochastic MLF-AG, particularly when the log-Sobolev constant $\alpha$ is small. Notably, it surpasses the conservative $\alpha = O(1/d)$ bound established in Proposition 3.2 of their paper.

## 4. Applications

### 4.1. Zero-Sum Games

Zero-sum games are widely applicable in economics, operations research, and reinforcement learning. These games involve a payoff function $G(\theta, \omega)$, which defines the interaction between two players' strategies $\theta$ and $\omega$. While finding a pure Nash equilibrium can be challenging or even impossible when $G$ is nonconvex-nonconcave, a mixed Nash equilibrium (MNE) often exists. In an MNE, players optimize their mixed strategies, represented as probability distributions over available actions.

Following Lu (2023), consider a bilinear distributional minimax optimization problem, where the strategies of two players are represented by probability distributions $\mu \in \mathcal{P}_1$ and $\nu \in \mathcal{P}_2$. The energy functional $J(\mu, \nu)$, which captures the expected payoff, is expressed as

$$J(\mu, \nu) = \mathbb{E}_{\mu\otimes\nu}[G(\theta, \omega)]. \tag{10}$$

The goal is to find an MNE $(\mu^*, \nu^*)$ that satisfies $\mu^* \in \arg\min_{\mu\in\mathcal{P}_1} E(\mu, \nu^*)$ and $\nu^* \in \arg\max_{\nu\in\mathcal{P}_2} E(\mu^*, \nu)$. This formulation extends the classical minimax problem to a distributional setting, where players optimize over probability measures rather than deterministic strategies.

**Example 1** (GAN). Consider the following generative adversarial network with an integral probability metric:

$$\min_{\mu\in\mathcal{P}(\mathbb{R}^{d_\Theta})} \max_{f\in\mathcal{F}} \left\{\mathbb{E}_\mu[f] - \mathbb{E}_{p_{\text{data}}}[f]\right\},$$

where $p_{\text{data}}$ is the real data distribution, $\mu$ is the distribution of the generated data, and $f$ is a discriminator function. Suppose that the discriminator function $f$ is parameterized by a (infinite-width) two-layer neural network with an activation function $\sigma(\cdot, \omega)$ parameterized by $\omega$ under the mean-field scaling, so that every function $f \in \mathcal{F}$ can be expressed as

$$f(\theta) = \mathbb{E}_{\omega\sim\nu}[\sigma(\theta, \omega)].$$

Then the generative adversarial network can be formulated as a distributional minimax optimization problem with

$$J(\mu, \nu) = \mathbb{E}_\mu\big[\mathbb{E}_\nu[\sigma(\theta, \omega)]\big] - \mathbb{E}_{p_{\text{data}}}\big[\mathbb{E}_\nu[\sigma(\theta, \omega)]\big],$$

which is a bilinear functional. ◇

The bilinear nature of the functional $J$ simplifies the analysis, as the Wasserstein gradients are given by

$$\nabla_\theta \frac{\delta J}{\delta \mu}[\mu, \nu] = \mathbb{E}_{\omega \sim \nu}[\nabla_\theta G(\theta, \omega)],$$

$$\nabla_\omega \frac{\delta J}{\delta \nu}[\mu, \nu] = \mathbb{E}_{\theta \sim \mu}[\nabla_\omega G(\theta, \omega)].$$

and the Gibbs distribution $\mathcal{K}_\mu^+[\nu]$ satisfies the first-order optimality condition of the inner maximization problem in (1). Under mild regularity conditions on $G$, we can verify that Assumptions 2-3 are satisfied.

**Proposition 3.** *Assume the payoff function $G$ satisfies that $\|\nabla^i G\|_F \le G_i$, $i = 0, 1, \ldots, 4$. Then the functional $J$ in (10) satisfies Assumption 2, and Assumption 3 holds with $\alpha = \frac{1}{\exp(2G_0 \tau^{-1})}$.*

### 4.2. Mean-Field Neural Networks

Consider a functional minimax problem

$$\min_f \max_g \ \mathbb{E}_{z \sim \mathcal{D}}[F(f(z), g(z), z)], \tag{11}$$

where $f$ and $g$ are functions of a variable $z$. The objective function $F$ is convex in $f$ and concave in $g$, and the expectation is taken with respect to $z \sim \mathcal{D}$. We parameterize $f$ and $g$ as infinite-width two-layer neural networks with activations $\sigma_1$ and $\sigma_2$, respectively:

$$f(z) = \mathbb{E}_{\theta \sim \mu}[\sigma_1(\theta, z)], \quad g(z) = \mathbb{E}_{\omega \sim \nu}[\sigma_2(\omega, z)].$$

This transforms the original problem into a minimax problem in distributional space:

$$\min_\mu \max_\nu \mathbb{E}_{z \sim \mathcal{D}}\big[F(\mathbb{E}_{\theta \sim \mu}[\sigma_1(z, \theta)], \mathbb{E}_{\omega \sim \nu}[\sigma_2(z, \omega)], z)\big]. \tag{12}$$

**Example 2** (Functional Conditional Moment Equations). The conditional moment equation is a fundamental problem in econometrics and statistics. Given a dataset $z = (X, Y) \sim \mathcal{D}$, the goal is to find a function $f$ that solves the following functional equation involving the conditional distribution of $X$ given $Z$:

$$\mathbb{E}_{Y|X}[\Phi(f(X, Y), Y) \mid X = x] = 0, \quad \forall x,$$

where $\Phi$ is a known function that is convex in $f$. Examples of $\Phi$ include conditional moment equations in nonparametric instrumental variable regression, policy evaluation in reinforcement learning, and asset pricing models in finance (Zhu et al., 2024). Using a Lagrangian dual function $g$, this problem can be formulated as a distributional minimax optimization problem by setting $g(z) = g(X)$ and

$$F(f, g, z) = g\Phi(f, Y). \qquad \diamond$$

**Example 3** (Feature-based Policy Learning). Feature-based decision-making (Yang et al., 2022) aims to find a policy $f$ from a set of features $X$ to an action $f(X)$. Given a data set $z = (X, Y) \sim \mathcal{D}$, where $Y$ is some exogenous random variable, the goal is to minimize the expected loss $\mathbb{E}_\mathcal{D}[\ell(f(X), Y)]$, subject to feasibility constraints $Af(X) \le b$ for every $X$. By introducing a Lagrangian dual function $g(z) = g(X)$, this problem can be formulated as a distributional minimax optimization problem with the objective

$$F(f, g, Z) = \ell(f, Z) + g(Af - b). \qquad \diamond$$

We introduce regularity assumptions on $F$ and $\sigma_1, \sigma_2$, which implies Assumption 2.

**Assumption 5.** The function $F(x, y)$ is convex-concave, $L$-smooth in both $x$ and $y$, and has bounded derivatives (i.e., $\|F_x'\|, \|F_y'\| \le F_1$). Moreover, we assume that $\sigma_1, \sigma_2$ has bounded gradients up to fourth-order, i.e., $\|\nabla^i \sigma_1\|, \|\nabla^i \sigma_2\| \le m_i, i = 0, 1, \ldots, 4$.

Under this regularity condition, we can also show that the Gibbs distributions $\mathcal{K}_\mu^+[\nu]$ and $\mathcal{K}_\nu^-[\mu]$ satisfy the log-Sobolev inequality, which verifies Assumption 3.

**Proposition 4.** *Under Assumption 5, (12) satisfies Assumption 2. Meanwhile, for $J(\mu, \nu)$ defined in (12), we have $\mathcal{K}_\mu^+[\nu] \in L^1(\mathbb{R}^{d_\Omega})$ and $\mathcal{K}_\nu^-[\mu] \in L^1(\mathbb{R}^{d_\Theta})$ and they both satisfy the Log-Sobolev inequality with parameter $\alpha_1 = \frac{1}{\exp(2F_1 m_0 \tau^{-1})}$.*

## 5. Conclusion

In this paper, we establish an $\tilde{O}(1/\epsilon)$ last-iterate convergence guarantee for the Mean-Field Langevin Stochastic Descent Ascent (MFL-SDA) algorithm. We also explore several common applications, including zero-sum games and mean-field neural networks.

There are several directions for future research. First, in practical applications, the MFL-SDA algorithm often requires finite-particle approximation, which calls for further analysis to establish uniform-in-time propagation of chaos. To address this issue, our analysis can be combined with the techniques from Chen et al. (2022); Suzuki et al. (2023; 2024); Nitanda (2024); Kim et al. (2023). Second, while our results apply to a broad class of nonlinear functionals, more specialized analyses for specific functionals–such as bilinear forms or convex-concave functions of expectations–may lead to sharper convergence guarantees. Lastly, our current algorithm follows a two-time-scale framework with a relatively large time-scale ratio, in line with Yang et al. (2020); Lu (2023). It remains an open question whether a single-timescale approach could be advantageous in certain settings, as highlighted in Wang & Chizat (2024).

## Impact Statement

This paper presents work whose goal is to advance the theoretical understanding of Machine Learning. There are many potential societal consequences of our work, none which we feel must be specifically highlighted here.

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

# A. Notation and Definition

## A.1. First-variation of functional

Let $J$ be a functional on $\mathcal{P}(\mathbb{R}^{d_\Theta}) \to \mathbb{R}$, its first-variation $\frac{\delta J}{\delta \mu}[\mu]$ at $\mu$ is defined as a functional $\mathcal{P}(\mathbb{R}^{d_\Theta}) \times \Theta \to \mathbb{R}$ satisfying for all $\nu \in \mathcal{P}(\mathbb{R}^{d_\Theta})$

$$\lim_{\epsilon \to 0} \frac{\mathrm{d}}{\mathrm{d}\epsilon} J(\mu + \epsilon(\nu - \mu)) = \int_\Theta \frac{\delta J}{\delta \mu}[\mu](\theta)(\nu - \mu)\mathrm{d}\theta. \tag{13}$$

The first-variation $\frac{\delta J}{\delta \mu}[\mu]$ is unique up to an additive constant, and for simplicity, we let $\int_\Theta \frac{\delta J}{\delta \mu}[\mu]\mathrm{d}\mu = J(\mu)$. For example, the first variation of $F(\mu) = \int_\Theta f \mathrm{d}\mu$ with respect to $\mu$ is exactly $f$.

The convexity of $J$ can also be expressed via its first variation, i.e. for all $\nu \in \mathcal{P}(\mathbb{R}^{d_\Theta})$:

$$J(\nu) \geq J(\mu) + \int_\Theta \frac{\delta J}{\delta \mu}[\mu](\theta)(\nu - \mu)(\mathrm{d}\theta). \tag{14}$$

And if $J$ is concave if $-J$ is convex.

## A.2. Tensor representation

For a function $f(x) : \mathbb{R}^d \to \mathbb{R}$, consider its Tayler expansion, we may denote the twice and third order derivative $\nabla^2 f(x) \in \mathbb{R}^{d \times d}$ and $\nabla^3 f(x) \in \mathbb{R}^{d \times d \times d}$ as s tensor. i.e.

$$(\nabla^2 f(x))_{ij} = \frac{\partial^2 f(x)}{\partial x_i \partial x_j}, \quad (\nabla^3 f(x))_{ijk} = \frac{\partial^3 f(x)}{\partial x_i \partial x_j \partial x_k}. \tag{15}$$

And the inner product with vector can be defined as

$$\langle \nabla^2 f(x), h(x)^{\otimes 2} \rangle = h(x)^\top \nabla^2 f(x) h(x), \quad \langle \nabla^3 f(x), h(x)^{\otimes 3} \rangle = \sum_{i,j,k} (\nabla^3 f(x))_{ijk} h(x)_i h(x)_j h(x)_k. \tag{16}$$

## A.3. Constants

In this section, we explicitly define the constants that have been used in the main content:

$$\begin{cases} \Gamma_0 = (\frac{M_2 + \tau}{2} r_{g4}^{1/2} + \frac{M_3}{6}(\eta_1 r_{g4}^{3/4} + 6\tau d r_{g4}^{1/2} + M_2^2 \tau/2 + 2d^2 M_4 \tau^2 + \frac{C_1 C_2}{\alpha^{1/2}\tau} r_{g4}^{1/2} \\ \quad + M_4(\eta_1^4 r_{g4} + 4\sqrt{2}\eta_1^{3.5}\tau^{1/2} m_1 r_{g4}^{3/4} + 12d\eta_1^3 \tau r_{g4}^{1/2} + 8\sqrt{2}m_3 \tau^{3/2}\eta_1^{5/2} r_{g4}^{1/2} + 4\eta_1^2 \tau^2 d(d+2)), \\ \Gamma_1 = (\frac{M_2 + \tau}{2} r_{g4}^{1/2} + \frac{M_3}{6}(\eta_2 r_{g4}^{3/4} + 6\tau d r_{g4}^{1/4}) + (M_2 + \tau)^2 \tau/2 + 2d^2 M_4 \tau^2 + \frac{C_1}{2} r_{g4}^{1/2} \\ \quad + M_4(\eta_2^4 r_{g4} + 4\sqrt{2}\eta_2^{3.5}\tau^{1/2} m_1 r_{g4}^{3/4} + 12d\eta_2^3 \tau r_{g4}^{1/2} + 8\sqrt{2}m_3 \tau^{3/2}\eta_2^{5/2} r_{g4}^{1/4} + 4\eta_2^2 \tau^2 d(d+2)), \\ \Gamma_2 = (\frac{M_2 + \tau}{2} r_{h4}^{1/2} + \frac{M_3}{6}(\eta_2 r_{h4}^{3/4} + 6\tau d r_{h4}^{1/4}) + (M_2 + \tau)^2 \tau/2 + 2d^2 M_4 \tau^2 + \frac{C_1}{2} r_{h4}^{1/2} \\ \quad + M_4(\eta_2^4 r_{h4} + 4\sqrt{2}\eta_2^{3.5}\tau^{1/2} m_1 r_{h4}^{3/4} + 12d\eta_2^3 \tau r_{h4}^{1/2} + 8\sqrt{2}m_3 \tau^{3/2}\eta_2^{5/2} r_{h4}^{1/4} + 4\eta_2^2 \tau^2 d(d+2)), \end{cases}$$

where constants $r_{g4}, r_{h4}$ are defined as

$$\begin{cases} r_{g4} = 8M_1^4 + 8\tau^3((2(4\tau + 2\tau d + \eta_1^3 M_1^2)^2/\tau + \eta_1^2 M_1^4 + (8 + 4d)\eta_1 \tau M_1^2 + 16\tau^2(d^2 + 2d))), \\ r_{h4} = 8M_1^4 + 8\tau^3((2(4\tau + 2\tau d + \eta_2^3 M_1^2)^2/\tau + \eta_2^2 M_1^4 + (8 + 4d)\eta_2 \tau M_1^2 + 16\tau^2(d^2 + 2d))). \end{cases}$$

The above constants are used in the proof of Theorem 1, and the following constants are used in the proof of Theorem 2:

$$
\begin{cases}
\Gamma_0 = (\dfrac{M_2 + \tau}{2} R_{g4}^{1/2} + \dfrac{M_3}{6}(\eta_1 R_{g4}^{3/4} + 6\tau d R_{g4}^{1/2} + M_2^2 \tau/2 + 2d^2 M_4 \tau^2 + \dfrac{C_1 C_2}{\alpha^{1/2}\tau} R_{g4}^{1/2} \\
\quad + M_4(\eta_1^4 R_{g4} + 4\sqrt{2}\eta_1^{3.5}\tau^{1/2} m_1 R_{g4}^{3/4} + 12d\eta_1^3 \tau R_{g4}^{1/2} + 8\sqrt{2}m_3 \tau^{3/2}\eta_1^{5/2} R_{g4}^{1/2} + 4\eta_1^2 \tau^2 d(d+2)), \\
\Gamma_1 = (\dfrac{M_2 + \tau}{2} R_{g4}^{1/2} + \dfrac{M_3}{6}(\eta_2 R_{g4}^{3/4} + 6\tau d R_{g4}^{1/4}) + (M_2 + \tau)^2 \tau/2 + 2d^2 M_4 \tau^2 + \dfrac{C_1}{2} R_{g4}^{1/2} \\
\quad + M_4(\eta_2^4 R_{g4} + 4\sqrt{2}\eta_2^{3.5}\tau^{1/2} m_1 R_{g4}^{3/4} + 12d\eta_2^3 \tau R_{g4}^{1/2} + 8\sqrt{2}m_3 \tau^{3/2}\eta_2^{5/2} R_{g4}^{1/4} + 4\eta_2^2 \tau^2 d(d+2)), \\
\Gamma_2 = (\dfrac{M_2 + \tau}{2} R_{h4}^{1/2} + \dfrac{M_3}{6}(\eta_2 R_{h4}^{3/4} + 6\tau d R_{h4}^{1/4}) + (M_2 + \tau)^2 \tau/2 + 2d^2 M_4 \tau^2 + \dfrac{C_1}{2} R_{h4}^{1/2} \\
\quad + M_4(\eta_2^4 R_{h4} + 4\sqrt{2}\eta_2^{3.5}\tau^{1/2} m_1 R_{h4}^{3/4} + 12d\eta_2^3 \tau R_{h4}^{1/2} + 8\sqrt{2}m_3 \tau^{3/2}\eta_2^{5/2} R_{h4}^{1/4} + 4\eta_2^2 \tau^2 d(d+2)),
\end{cases}
\tag{17}
$$

where constants $R_{g4}, R_{h4}$ are defined as

$$
\begin{cases}
R_{g4} = 8(M_1^4 + \zeta) + 8\tau^3((2(4\tau + 2\tau d + \eta_1^3(M_1^2 + \zeta^{1/2}))^2/\tau \\
\quad + \eta_1^2(M_1^2 + \zeta^{1/2}) + (8 + 4d)\eta_1 \tau(M_1^2 + \zeta^{1/2}) + 16\tau^2(d^2 + 2d))), \\
R_{h4} = 8(M_1^4 + \zeta) + 8\tau^3((2(4\tau + 2\tau d + \eta_2^3(M_1^2 + \zeta^{1/2}))^2/\tau \\
\quad + \eta_2^2(M_1^2 + \zeta^{1/2}) + (8 + 4d)\eta_2 \tau(M_1^2 + \zeta^{1/2}) + 16\tau^2(d^2 + 2d))).
\end{cases}
$$

# B. Preliminary Results

## B.1. Log-Sobolev Inequality

In this section, we present some sufficient conditions for ensuring log-Sobolev inequality from the existing literature.

**Lemma 3** (Bakry & Émery (2006)). *If $f : \mathbb{R}^d \to \mathbb{R}$ is a function and $\nabla^2 f \succeq \alpha I$, then the probability density $\rho \propto \exp(-f)$ satisfies LSI with constant $\alpha$.*

**Lemma 4** (Holley & Stroock (1986)). *Let $\rho$ be a density satisfying LSI with constant $\alpha$, then for a bounded function $B : \mathbb{R}^d \to \mathbb{R}$, the perturbed distribution*

$$
dp_B(x) = \frac{\exp(B(x))\rho(x)}{\mathbb{E}_\rho[\exp(B(x)]}dx,
\tag{18}
$$

*satisfies LSI with parameter $\exp(-(\sup B - \inf B))\alpha$.*

**Lemma 5** (Suzuki et al. (2024); Kim et al. (2023)). *Let the probability measure $\mu \propto \exp(-\tau^{-1}h - \frac{\|\theta\|_2^2}{2})$ with $\|\nabla h\| \leq M_1$, then $\mu$ satisfies LSI with constant*

$$
\alpha \geq \frac{1}{2}e^{-\frac{4M_1^2}{\tau^2}}\sqrt{\frac{2d}{\pi}} \vee \left(4 + \left(\frac{M_1}{\tau} + \sqrt{2}\right)^2 \left(2 + d + \frac{4M_1^2}{\tau^2}\right)e^{\frac{M_1^2}{2\tau^2}}\right)^{-1}.
$$

## B.2. Technical Lemmas

**Lemma 6.** *Let $\mu$ be a probability density and let $\rho_\epsilon$ be its convolution with a normal distribution $N(0, \epsilon I)$, then*

$$
\mathbb{E}_\rho[\log \rho_\epsilon] \leq \mathbb{E}_\rho[\log \rho] - \frac{\epsilon}{2}I(\rho) + C\epsilon^2,
$$

*where $I(\rho)$ is Fisher information of $\rho$, and $C$ is a uniform bound on the Hessian of $\log \rho$.*

*Proof of Lemma 6.* Define $h : \mathbb{R}_+ \to \mathbb{R}$ as

$$
h(\epsilon) = \mathbb{E}_{\rho_\epsilon}[\log \rho_\epsilon].
$$

Then by de Brujin's identity (Stam, 1959), we have $h'(\epsilon) = -\frac{1}{2}I(\rho_\epsilon) \leq 0$. Based on the data processing inequality, $I(\rho_\epsilon)$ is nonincreasing in $\epsilon$, and hence $h'$ is nondecreasing. This establishes the convexity of $h$, and we arrive at

$$
h(\epsilon) \leq h(0) + \epsilon h'(0) + C\epsilon^2,
$$

which yields the desired result. □

**Lemma 7.** *For measures $\nu_{k+1}, \nu_k$ defined in Algorithm 1, we have*

$$\tau\mathcal{H}(\nu_{k+1}) - \tau\mathcal{H}(\nu_k) \geq \eta_2\mathbb{E}_{\nu_k}[\|\tau\nabla\log\nu_k\|_2^2] - \eta_2\mathbb{E}_{\nu_k}[\langle h_k, \tau\nabla\log\nu_k\rangle] - \tilde{O}(\tau\eta_2^2). \tag{19}$$

*Similarly, for $\mu_{k+1}, \mu_k$, we have*

$$\tau\mathcal{H}(\mu_k) - \tau\mathcal{H}(\mu_{k+1}) \leq -\eta_2\mathbb{E}_{\mu_k}[\|\tau\nabla\log\mu_k\|_2^2] - \eta_2\mathbb{E}_{\mu_k}[\langle g_k, \tau\nabla\log\mu_k\rangle] + \tilde{O}(\tau\eta_1^2). \tag{20}$$

*where $\tilde{O}(\tau\eta_i^2) = (M_2 + \tau)^2\tau\eta_i^2/2 + 2d^2M_4\tau^2\eta_i^2 + C\tau\eta_i^2$.*

*Proof of Lemma 7.* We first prove (19). Let $\bar{\nu}_{k+1}$ be the distribution of $\bar{\omega}_{k+1} = T(\omega_k)$ with $T(\omega_k) = \omega_k + \eta_2 h_k$ and $\omega_k \sim \nu_k$. With this definition, we can decompose $\mathcal{H}(\mu_k) - \mathcal{H}(\mu_{k+1})$ into the sum of the following two terms

$$-\mathbb{E}_{\bar{\nu}_{k+1}}[\log\bar{\nu}_{k+1}] + \mathbb{E}_{\nu_k}[\log\nu_k] \quad \text{and} \quad \mathbb{E}_{\bar{\nu}_{k+1}}[\log\bar{\nu}_{k+1}] - \mathbb{E}_{\nu_{k+1}}[\log\nu_{k+1}]. \tag{21}$$

Note that $T$ is invertible for $0 < \eta_2 \leq 1/M_2$. By the formula for change of variables, we arrive at

$$\begin{aligned}
\mathbb{E}_{\bar{\nu}_{k+1}}[\log\bar{\nu}_{k+1}] &= \int \bar{\nu}_{k+1}(\bar{\omega}_{k+1})\log\bar{\nu}_{k+1}(\bar{\omega}_{k+1})\mathrm{d}\bar{\omega}_{k+1} \\
&= \int \bar{\nu}_{k+1}(T(\omega_k))\log\bar{\nu}_{k+1}(T(\omega_k))|\det(\nabla T(\omega_k))|\mathrm{d}\omega_k \\
&= \int \nu_k(\omega_k)\log\nu_k(\omega_k)\mathrm{d}\omega_k - \int \nu_k(\omega_k)\log|\det(\nabla T(\omega_k))|\mathrm{d}\omega_k \\
&= \mathbb{E}_{\nu_k}[\log\nu_k] - \mathbb{E}_{\nu_k}[\log|\det(\nabla T)|].
\end{aligned} \tag{22}$$

Hence, the first term in (21) becomes

$$\begin{aligned}
-\mathbb{E}_{\bar{\nu}_{k+1}}[\log\bar{\nu}_{k+1}] + \mathbb{E}_{\nu_k}[\log\nu_k] &= \mathbb{E}_{\nu_k}[\log|\det(\nabla T)|] \\
&= \mathbb{E}_{\nu_k}[\log\det(I + \eta_2\nabla h_k(\omega_k))] \\
&= \mathbb{E}_{\nu_k}[\mathrm{Tr}(\eta_2\nabla h_k(\omega_k) + \eta_2^2(\nabla h_k(\bar{\omega}_k))^2/2)] \\
&\geq \eta_2\mathbb{E}_{\nu_k}[\nabla \cdot h_k] - (M_2 + \tau)^2\eta_2^2/2 \\
&= -\eta_2\mathbb{E}_{\nu_k}[\langle h_k, \nabla\log\nu_k\rangle] - (M_2 + \tau)^2\eta_2^2/2.
\end{aligned} \tag{23}$$

Here, the last equality follows since

$$= \mathbb{E}_{\nu_k}[\nabla \cdot h_k] = \int_\Omega \nu_k\nabla \cdot h_k\mathrm{d}\omega = -\int_\Omega h_k \cdot \nabla\nu_k\mathrm{d}\omega = -\mathbb{E}_{\nu_k}[\langle h_k, \nabla\log\nu_k\rangle]. \tag{24}$$

Note that $\nu_{k+1}$ equals the convolution of $\bar{\nu}_{k+1}$ and a Gaussian distribution $N(0, 2\eta_2\tau I)$. By Lemma 7, the second term in (21) can be bounded by

$$\tau(\mathbb{E}_{\bar{\nu}_{k+1}}[\log\bar{\nu}_{k+1}] - \mathbb{E}_{\nu_{k+1}}[\log\nu_{k+1}]) \geq \eta_2\mathbb{E}_{\nu_k}[\|\tau\nabla\log\nu_k\|_2^2] - \eta_2(\mathbb{E}_{\nu_k}[\|\tau\nabla\log\nu_k\|_2^2] - \mathbb{E}_{\bar{\nu}_{k+1}}[\|\tau\nabla\log\bar{\nu}_{k+1}\|_2^2]), \tag{25}$$

where the last term on the right-hand side can be further bounded by

$$\begin{aligned}
\mathbb{E}_{\bar{\nu}_{k+1}}[\|\nabla\log\bar{\nu}_{k+1}\|_2^2] &= \int \bar{\nu}_{k+1}\|\nabla\log\bar{\nu}_{k+1}\|_2^2\mathrm{d}\bar{\omega}_{k+1} \\
&= \int \nu_k\|\nabla\log\nu_k(\omega_k) - \nabla\log|\det(\nabla T(\omega_k))|\|_2^2\mathrm{d}\omega_k \\
&\geq \mathbb{E}_{\nu_k}[\|\nabla\log\nu_k\|_2^2] - 2\int \nabla\log\nu_k^\top\nabla\log|\det(\nabla T(\omega_k))|\nu_k(\omega_k)\mathrm{d}\omega_k \\
&\geq \mathbb{E}_{\nu_k}[\|\nabla\log\nu_k\|_2^2] - 2\eta_2\sup_{\omega_k}\Delta^2 g_k(\omega_k).
\end{aligned} \tag{26}$$

Here, the last inequality follows since

$$\left| \int \nabla \nu_k(\omega_k)^\top \nabla \log |\det(\nabla T(\omega_k))| \mathrm{d}\omega_k \right| = \left| \int \nu_k(\omega_k) \Delta(\log |\det(\nabla T(\omega_k))|) \mathrm{d}\omega_k \right|$$

$$= \int \nu_k(\omega_k) \Delta(\mathrm{tr}(\eta_2 \nabla \cdot h_k(\tilde{\omega}_k)) \mathrm{d}\omega_k \tag{27}$$

$$= \eta_2 \int \nu_k(\omega_k) \Delta^2 \frac{\delta J}{\delta \nu}[\mu_{k+1}, \nu_k](\tilde{\omega}_k) \mathrm{d}\omega_k.$$

Since $\Delta^2 h_k < d^2 M_4$, it holds that

$$\eta_2(\mathbb{E}_{\nu_k}[\|\tau \nabla \log \nu_k\|_2^2] - \mathbb{E}_{\bar{\nu}_{k+1}}[\|\tau \nabla \log \bar{\nu}_{k+1}\|_2^2]) \le 2d^2 M_4 \tau^2 \eta_2^2. \tag{28}$$

Combining these terms together we then obtain the desired result (19). Finally, note that (19) and (20) are symmetric. Hence, (20) can be proved using a similar technique and we omit here for brevity. $\square$

**Lemma 8.** *For any $\mu \in \mathcal{P}(\mathbb{R}^{d_\Theta})$ and $\nu \in \mathcal{P}(\mathbb{R}^{d_\Omega})$, the following inqualities hold:*

$$\tau \mathsf{KL}(\mu|\mu^*) \le \mathcal{L}_1(\mu) \le \tau \mathsf{KL}(\mu|\mathcal{K}_\mu^-[\mathcal{K}_*^+[\mu]]),$$
$$\tau \mathsf{KL}(\nu|\mathcal{K}_*^+[\mu]) \le \mathcal{L}_2(\mu, \nu) \le \tau \mathsf{KL}(\nu|\mathcal{K}_\nu^+[\mu]).$$

*Proof of Lemma 8.* We start by establishing the lower bound and upper bound of $\mathcal{L}_2$ via KL-divergence. Specifically,

$$\mathcal{L}_2(\mu, \nu) = \tau \mathsf{KL}(\nu|\rho^\nu) - \tau \mathsf{KL}(\mathcal{K}_*^+[\mu]|\rho^\nu) + J(\mu, \mathcal{K}_*^+[\mu]) - J(\mu, \nu)$$

$$\ge \tau \mathsf{KL}(\nu|\rho^\nu) - \tau \mathsf{KL}(\mathcal{K}_*^+[\mu]|\rho^\nu) - \int \frac{\delta J}{\delta \nu}[\mu, \mathcal{K}_*^+[\mu]] \mathrm{d}(\nu - \mathcal{K}_*^+[\mu])$$

$$\ge \tau \mathsf{KL}(\nu|\mathcal{K}_*^+[\mu]),$$

$$\mathcal{L}_2(\mu, \nu) = \tau \mathsf{KL}(\nu|\rho^\nu) - \tau \mathsf{KL}(\mathcal{K}_*^+[\mu]|\rho^\nu) + J(\mu, \mathcal{K}_*^+[\mu]) - J(\mu, \nu) \tag{29}$$

$$\le \tau \mathsf{KL}(\nu|\rho^\nu) - \tau \mathsf{KL}(\mathcal{K}_*^+[\mu]|\rho^\nu) + \int \frac{\delta J}{\delta \nu}[\mu, \nu] \mathrm{d}(\mathcal{K}_*^+[\mu] - \nu)$$

$$\le \tau \mathsf{KL}(\nu|\mathcal{K}_\nu^+[\mu])).$$

The inequalities is a consequence of convex-concave property of functioinal $J$, see Appendix A.1. To prove inequalities of $\mathcal{L}_1(\mu)$, we first need a symmetric functional of $\mathcal{L}_1, \mathcal{L}_2$, named $\mathcal{L}_3, \mathcal{L}_4$ defined by

$$\mathcal{L}_3(\nu) := \max_{\nu' \in \mathcal{P}(\mathbb{R}^{d_\Omega})} \min_{\mu' \in \mathcal{P}(\mathbb{R}^{d_\Theta})} E(\mu', \nu') - \min_{\mu' \in \mathcal{P}(\mathbb{R}^{d_\Theta})} E(\mu', \nu),$$

$$\mathcal{L}_4(\mu, \nu) := E(\mu, \nu) - \min_{\mu' \in \mathcal{P}(\mathbb{R}^{d_\Theta})} E(\mu', \nu). \tag{30}$$

By the same technique as in (29), we can verify that

$$\mathcal{L}_4(\mu, \nu) \le \tau \mathsf{KL}(\mu|\mathcal{K}_\mu^-[\nu]). \tag{31}$$

Then we arrive at

$$\mathcal{L}_1(\mu) = \max_{\nu' \in \mathcal{P}(\mathbb{R}^{d_\Omega})} E(\mu, \nu') - \min_{\mu' \in \mathcal{P}(\mathbb{R}^{d_\Theta})} \max_{\nu' \in \mathcal{P}(\mathbb{R}^{d_\Omega})} E(\mu', \nu')$$

$$= \max_{\nu' \in \mathcal{P}(\mathbb{R}^{d_\Omega})} E(\mu, \nu') - \min_{\mu' \in \mathcal{P}(\mathbb{R}^{d_\Theta})} E(\mu', \mathcal{K}_*^+[\mu])$$

$$\le E(\mu, \mathcal{K}_*^+[\mu]) - \min_{\mu' \in \mathcal{P}(\mathbb{R}^{d_\Theta})} E(\mu', \mathcal{K}_*^+[\mu]) \tag{32}$$

$$= \mathcal{L}_4(\mu, \mathcal{K}_*^+[\mu])$$

$$\le \tau \mathsf{KL}(\mu|\mathcal{K}_\mu^-[\mathcal{K}_*^+[\mu]]).$$

The inequality is since we fixed the value of second term in $E$. It remains to establish a lower bound for $\mathcal{L}_1(\mu)$. To this end, we note that

$$\log \mathcal{Z}^*[\alpha\mu_1 + (1-\alpha)\mu_2]$$
$$= \log \left( \int \exp\left( \alpha \int \tau^{-1} \frac{\delta}{\delta\nu} J[\mu_1, \mathcal{K}_*^+[\mu_1]] + \|\omega\|_2^2 \mathrm{d}\mu \right) \cdot \exp\left( (1-\alpha) \int \tau^{-1} \frac{\delta}{\delta\nu} J[\mu_2, \mathcal{K}_*^+[\mu_2]] + \|\omega\|_2^2 \mathrm{d}\mu \right) \mathrm{d}\omega \right)$$
$$\leq \log \left( \left( \int \exp\left( \int \tau^{-1} \frac{\delta}{\delta\nu} J[\mu_1, \mathcal{K}_*^+[\mu_1]] + \|\omega\|_2^2 \mathrm{d}\mu \right) \mathrm{d}\omega \right)^\alpha \cdot \left( \int \exp\left( \int \tau^{-1} \frac{\delta}{\delta\nu} J[\mu_2, \mathcal{K}_*^+[\mu_2]] + \|\omega\|_2^2 \mathrm{d}\mu \right) \mathrm{d}\omega \right)^{1-\alpha} \right)$$
$$= \alpha \log \mathcal{Z}^*[\mu_1] + (1-\alpha) \log \mathcal{Z}^*[\mu_2],$$

which verifies the convexity of $\log \mathcal{Z}^*$. Then we obtain

$$\begin{aligned}
\mathcal{L}_1(\mu) &= E^*(\mu) - E^*(\mu^*) \\
&= -\tau \left( \mathcal{H}(\mu) - \mathcal{H}(\mu^*) \right) + \tau(\log \mathcal{Z}^*[\mu] - \log \mathcal{Z}^*[\mu^*]) \\
&\geq \tau \left( \int \log\mu \mathrm{d}\mu - \int \log\mu^* \mathrm{d}\mu^* \right) - \tau \int \frac{\delta \log \mathcal{Z}^*[\mu^*]}{\delta\mu} \mathrm{d}(\mu - \mu^*) \\
&= \tau \left( \int \log\mu \mathrm{d}\mu - \int \log\mu^* \mathrm{d}\mu^* \right) - \tau \int \log(\mu^*) \mathrm{d}(\mu - \mu^*) \\
&= \tau\mathsf{KL}(\mu|\mu^*),
\end{aligned} \tag{33}$$

where the first inequality follows from the convexity of $\log \mathcal{Z}^*$. Hence, we complete the proof. $\qquad\square$

**Lemma 9.** *Given $\mu, \mu' \in \mathcal{P}(\mathbb{R}^{d_\Theta})$, we can obtain*

$$\mathsf{KL}(\mathcal{K}_*^+[\mu']|\mathcal{K}_*^+[\mu]) \leq \frac{1}{2\alpha\tau^2} \mathbb{E}_{\mathcal{K}_*^+[\mu']} \left[ \left\| \int \nabla_\omega \frac{\delta^2 J}{\delta\mu\delta\nu} [\mu, \mathcal{K}_*^+[\tilde{\mu}]] \mathrm{d}(\mu' - \mu) \right\|_2^2 \right]. \tag{34}$$

*Proof of Lemma 9.* First, consider

$$\mathcal{L}_2(\mu, \mathcal{K}_*^+[\mu']) = E(\mu, \mathcal{K}_*^+[\mu]) - E(\mu, \mathcal{K}_*^+[\mu']).$$

By Lemma 8 we obtain

$$\mathcal{L}_2(\mu, \mathcal{K}_*^+[\mu']) \geq \tau\mathsf{KL}(\mathcal{K}_*^+[\mu']|\mathcal{K}_*^+[\mu]).$$

The last inequality follows from the fact that if $\nu$ satisfies (LSI) with constant $\alpha$, then it satisfies the Talagrand inequality. Also, we use another side of Lemma 8 we can get

$$\begin{aligned}
\mathcal{L}_2(\mu, \mathcal{K}_*^+[\mu']) &\leq \tau\mathsf{KL}(\mathcal{K}_*^+[\mu']|\mathcal{K}_\mu[\mathcal{K}_*^+[\mu']]) \\
&\leq \frac{\tau}{2\alpha} \mathbb{E}_{\mathcal{K}_*^+[\mu']} \left[ \left\| \nabla_\omega \log \frac{\mathcal{K}_*^+[\mu']}{\mathcal{K}_\mu[\mathcal{K}_*^+[\mu']]} \right\|_2^2 \right] \\
&= \frac{1}{2\alpha\tau} \mathbb{E}_{\mathcal{K}_*^+[\mu']} \left[ \left\| \nabla_\omega \frac{\delta J}{\delta\nu} [\mu', \mathcal{K}_*^+[\mu'] - \nabla_\omega \frac{\delta J}{\delta\nu} [\mu, \mathcal{K}_*^+[\mu'] \right\|_2^2 \right] \\
&= \frac{1}{2\alpha\tau} \mathbb{E}_{\mathcal{K}_*^+[\mu']} \left[ \left\| \int \nabla_\omega \frac{\delta^2 J}{\delta\mu\delta\nu} [\tilde{\mu}, \mathcal{K}_*^+[\mu'] \mathrm{d}(\mu' - \mu) \right\|_2^2 \right]
\end{aligned} \tag{35}$$

Combing the above two equation we can get desired result. $\qquad\square$

**Lemma 10.** *Given Assumptions (2)-(3), we can prove under the algorithm 1,*

$$\mathbb{E}_{\nu_k}[\|h_k\|_2^4] \leq \mathbb{E}_{\nu_0}[\|\omega_0\|_2^4] + r_{h4}, \qquad \mathbb{E}_{\mu_k}[\|g_k\|_2^4] \leq \mathbb{E}_{\mu_0}[\|\theta_0\|_2^3] + r_{g4}. \tag{36}$$

*where $r_{h4}, r_{g4}$ are specified in the proof.*

*Proof of Lemma 10.* Let $\rho = \mathcal{N}(0, I_d)$ be a standard multivariate Gaussian distribution. We first consider the second-order moment, note that

$$
\begin{aligned}
&\mathbb{E}_{\nu_k \otimes \rho}[\|\omega_k + \eta_2 h_k\|_2^2] \\
=&\mathbb{E}_{\nu_k}[\|\omega_k\|_2^2] + 2\mathbb{E}_{\nu_k \otimes \rho}[\langle \omega_k, \eta_2(h_k + \tau\omega_k - \tau\omega_k)\rangle] + \mathbb{E}_{\nu_k \otimes \rho}[\|\eta_2(h_k + \tau\omega_k - \tau\omega_k)\|_2^2] \\
\leq&\mathbb{E}_{\nu_k}[\|\omega_k\|_2^2] + 2\eta_2 M_1 \mathbb{E}_{\nu_k}[\|\omega_k\|] - 2\eta_2\tau\mathbb{E}_{\nu_k}[\|\omega_k\|_2^2] + 2\eta_2^2(M_1^2 + \tau^2\mathbb{E}_{\nu_k}[\|\omega_k\|_2^2]) \\
\leq&(1 - 2\eta_2\tau + \frac{\eta_2\tau}{2} + 2\eta_2^2\tau^2)\mathbb{E}_{\nu_k}[\|\omega_k\|_2^2] + 2\eta_2 M_1^2/\tau + 2\eta_2^2 M_1^2 \\
\leq&(1 - \eta_2\tau)\mathbb{E}_{\nu_k}[\|\omega_k\|_2^2] + \eta_2(2M_1^2/\tau + 2\eta_2 M_1^2),
\end{aligned}
\tag{37}
$$

where the first inequality follows the fact that $\|h_k + \tau\omega_k\| = \|\nabla\frac{\delta J}{\delta\nu}[\mu_{k+1}, \nu_k](\omega_k)\| \leq M_1$, the second inequality is since $2M_1\mathbb{E}_{\nu_k}[\|\omega_k\|_2] \leq \frac{\tau}{2}(\mathbb{E}_{\nu_k}[\|\omega_k\|_2])^2 + 2M_1^2/\tau \leq \frac{\tau}{2}\mathbb{E}_{\nu_k}[\|\omega_k\|_2^2] + 2M_1^2/\tau$ and last inequality since $\eta_2\tau < 1/4$.

Then we consider the fourth moment,

$$
\begin{aligned}
&\mathbb{E}_{\nu_{k+1}}[\|\omega_{k+1}\|_2^4] \\
=&\mathbb{E}_{\nu_k \otimes \rho}[\|\omega_k + \eta_2 h_k + \sqrt{2\eta_2\tau}\xi_k^2\|_2^4] \\
\leq&\mathbb{E}_{\nu_k \otimes \rho}\left[\|\omega_k + \eta_2 h_k\|^4 + (8 + 4d)\eta_2\tau\|\omega_k + \eta_2 h_k\|_2^2 + 4\eta_2^2\tau^2(d^2 + 2d)\right].
\end{aligned}
\tag{38}
$$

Since we have prove

$$
\mathbb{E}_{\nu_k}[\|\omega_k + \eta_2 h_k\|_2^2] \leq (1 - \eta_2\tau)\mathbb{E}_{\nu_k}[\|\omega_k\|_2^2] + \eta_2(2M_1^2/\tau + 2\eta_2 M_1^2).
$$

Then

$$
\begin{aligned}
\mathbb{E}_{\nu_k}[\|\omega_k + \eta_2 h_k\|_2^4] &\leq \mathbb{E}_{\nu_k}[((1 - 3/2\eta_2\tau + \eta_2^2\tau^2)\mathbb{E}_{\nu_k}[\|\omega_k\|_2^2] + \eta_2(2M_1^2/\tau + 2\eta_2 M_1^2))^2] \\
&= (1 - 3/2\eta_2\tau + \eta_2^2\tau^2)^2\mathbb{E}_{\nu_k}[\|\omega_k\|_2^4] + 2\eta_2^4 M_1^2(1 - 3/2\eta_2\tau + \eta_2^2\tau^2)\mathbb{E}_{\nu_k}[\|\omega_k\|_2^2] + \eta_2^4(2M_1^2/\tau + 2\eta_2 M_1^2))^2.
\end{aligned}
$$

Then combining them together,

$$
\begin{aligned}
&\mathbb{E}_{\nu_k}[\|\omega_k\|_2^4] \\
\leq&(1 - 3/2\eta_2\tau + \eta_2^2\tau^2)^2\mathbb{E}_{\nu_k}[\|\omega_k\|_2^4] + \eta_2(1 - 3/2\eta_2\tau + \eta_2^2\tau^2)(8\tau + 4\tau d + 2\eta_2^3 M_1^2)\mathbb{E}_{\nu_k}[\|\omega_k\|_2^2] \\
&+\eta_2^4 M_1^4 + (8 + 4d)\eta_2^3\tau M_1^2 + 16\eta_2^2\tau^2(d^2 + 2d) \\
\leq&(1 - \eta_2\tau)\mathbb{E}_{\nu_k}[\|\omega_k\|_2^4] + \eta_2(2(4\tau + 2\tau d + \eta_2^3 M_1^2)^2/\tau + \eta_2^2 M_1^4 + (8 + 4d)\eta_2\tau M_1^2 + 16\tau^2(d^2 + 2d))
\end{aligned}
\tag{39}
$$

Hence, by iteratively applying this inequality we can get

$$
\mathbb{E}_{\nu_k}[\|\omega_k\|_2^4] \leq \mathbb{E}_{\nu_0}[\|\omega_0\|_2^4] + \frac{1}{\eta_2\tau}\eta_2(2(4\tau + 2\tau d + \eta_2^3 M_1^2)^2/\tau + \eta_2^2 M_1^4 + (8 + 4d)\eta_2\tau M_1^2 + 16\tau^2(d^2 + 2d)).
$$

Then by $(a + b)^4 \leq 8(a^4 + b^4)$ we can obtain the upper bound of fourth moment of gradient $h_k$:

$$
\begin{aligned}
&\mathbb{E}_{\nu_k}[\|h_k\|_2^4] \leq 8(\mathbb{E}_{\nu_k}[\|h_k - \tau\omega_k\|_2^4 + \|\tau\omega_k\|_2^4] \\
\leq&8M_1^4 + 8\tau^3(\mathbb{E}_{\nu_k}[\|\omega_0\|_2^4] + (2(4\tau + 2\tau d + \eta_2^3 M_1^2)^2/\tau + \eta_2^2 M_1^4 + (8 + 4d)\eta_2\tau M_1^2 + 16\tau^2(d^2 + 2d))).
\end{aligned}
$$

Denote $r_{h4} = 8M_1^4 + 8\tau^3((2(4\tau + 2\tau d + \eta_2^3 M_1^2)^2/\tau + \eta_2^2 M_1^4 + (8 + 4d)\eta_2\tau M_1^2 + 16\tau^2(d^2 + 2d)))$.

For $\mathbb{E}_{\mu_k}[\|g_k\|_2^4]$, we can similarly obtain

$$
\mathbb{E}_{\mu_k}[\|g_k\|_2^4] \leq \mathbb{E}_{\mu_0}[\|\theta_0\|_2^4] + r_{g4},
$$

where $r_{h4} = 8M_1^4 + 8\tau^3((2(4\tau + 2\tau d + \eta_1^3 M_1^2)^2/\tau + \eta_1^2 M_1^4 + (8 + 4d)\eta_1\tau M_1^2 + 16\tau^2(d^2 + 2d)))$.

$\square$

## C. Proofs for Section 3

*Proof of Proposition 1.* To prove the boundedness of $E(\mu_{k+1}, \mathcal{K}_*^+[\mu_{k+1}]) - E(\mu_k, \mathcal{K}_*^+[\mu_k])$, we split it into

$$\left(E(\mu_{k+1}, \mathcal{K}_*^+[\mu_{k+1}]) - E(\mu_{k+1}, \mathcal{K}_*^+[\mu_k])\right) + \left(E(\mu_{k+1}, \mathcal{K}_*^+[\mu_k]) - E(\mu_k, \mathcal{K}_*^+[\mu_k])\right). \tag{40}$$

The second term includes

$$E(\mu_{k+1}, \mathcal{K}_*^+[\mu_k]) - E(\mu_k, \mathcal{K}_*^+[\mu_k]) = J(\mu_{k+1}, \mathcal{K}_*^+[\mu_k]) - J(\mu_k, \mathcal{K}_*^+[\mu_k]) + \tau\left(\mathbb{E}_{\mu_{k+1}}[\log \mu_{k+1}] - \mathbb{E}_{\mu_k}[\log \mu_k]\right)$$
$$+ \frac{\tau}{2}\left(\mathbb{E}_{\mu_{k+1}}[\|\theta_{k+1}\|_2^2] - \mathbb{E}_{\mu_k}[\|\theta_k\|_2^2]\right) \tag{41}$$

Where the entropy term has bound

$$\tau\left(\mathbb{E}_{\mu_{k+1}}[\log \mu_{k+1}] - \mathbb{E}_{\mu_k}[\log \mu_k]\right) \leq \eta_2 \mathbb{E}_{\mu_k}[\|\tau \nabla \log \mu_k\|_2^2] - \eta_2 \mathbb{E}_{\mu_k}[\langle g_k, \tau \nabla \log \mu_k \rangle] + (M_2 + \tau)^2 \tau \eta_1^2/2 + 2d^2 M_4 \tau^2 \eta_1^2.$$

via Lemma 7. Now we consider the weight decay term $\frac{\tau}{2}\left(\mathbb{E}_{\mu_{k+1}}[\|\theta_{k+1}\|_2^2] - \mathbb{E}_{\mu_k}[\|\theta_k\|_2^2]\right)$. Let $\rho = \mathcal{N}(0, I_d)$ be a standard multivariate Gaussian distribution, then:

$$\frac{\tau}{2}\mathbb{E}_{\mu_{k+1} - \mu_k}[\|\theta\|_2^2] = \frac{\tau}{2}\mathbb{E}_{\mu_k \otimes \rho}[\|\theta_k + \eta_1 g_k + \sqrt{2\eta_1 \tau}\xi_k^1\|_2^2 - \|\theta_k\|_2^2]$$
$$= \frac{\tau}{2}\mathbb{E}_{\mu_k}[2\eta_1 \langle \theta_k, g_k \rangle + \eta_1^2 \|g_k\|_2^2] + \eta_1 \tau^2 d. \tag{42}$$

And by second-order expansion of $J(\cdot, \nu)$,

$$J(\mu_{k+1}, \mathcal{K}_*^+[\mu_k]) - J(\mu_k, \mathcal{K}_*^+[\mu_k])$$
$$\leq \mathbb{E}_{\mu_{k+1} - \mu_k}\left[\frac{\delta J}{\delta \mu}[\mu_k, \mathcal{K}_*^+[\mu_k]](\theta_k)\right] + \frac{1}{2}\int_{\mathbb{R}^d} \frac{\delta^2 J}{\delta \mu^2}[\tilde{\mu}, \mathcal{K}_*^+[\mu_k]](\theta, \theta')\mathrm{d}(\mu_{k+1} - \mu_k)(\theta)\mathrm{d}(\mu_{k+1} - \mu_k)(\theta') \tag{43}$$

By the fourth-order Taylor expansion with Lagrangian remainder, the first part on the right-hand side of (43) becomes

$$\mathbb{E}_{\mu_{k+1} - \mu_k}\left[\frac{\delta J}{\delta \mu}[\mu_k, \mathcal{K}_*^+[\mu_k]]\right] \tag{44}$$

$$= \mathbb{E}_{\mu_k \otimes \rho}\left[\frac{\delta J}{\delta \mu}[\mu_k, \mathcal{K}_*^+[\mu_k]](\theta_k - \eta_1 g_k + \sqrt{2\eta_1 \tau}\xi_k^1) - \frac{\delta J}{\delta \mu}[\mu_k, \mathcal{K}_*^+[\mu_k]](\theta_k)\right]$$

$$= \mathbb{E}_{\mu_k \otimes \rho}\left[\left\langle \nabla \frac{\delta J}{\delta \mu}[\mu_k, \mathcal{K}_*^+[\mu_k]], -\eta_1 g_k + \sqrt{2\eta_1 \tau}\xi_k^1 \right\rangle + \frac{1}{2}\left\langle \nabla^2 \frac{\delta J}{\delta \mu}[\mu_k, \mathcal{K}_*^+[\mu_k]], (-\eta_1 g_k + \sqrt{2\eta_1 \tau}\xi_k^1)^{\otimes 2} \right\rangle\right] \tag{45}$$

$$+ \frac{1}{6}\mathbb{E}_{\mu_k \otimes \rho}\left[\left\langle \nabla^3 \frac{\delta J}{\delta \mu}[\mu_k, \mathcal{K}_*^+[\mu_k]], (-\eta_1 g_k + \sqrt{2\eta_1 \tau}\xi_k^1)^{\otimes 3} \right\rangle\right], \tag{46}$$

$$+ \frac{1}{24}\mathbb{E}_{\mu_k \otimes \rho}\left[\left\langle \nabla^4 \frac{\delta J}{\delta \mu}[\mu_k, \mathcal{K}_*^+[\mu_k]](\tilde{\theta}_k), (-\eta_1 g_k + \sqrt{2\eta_1 \tau}\xi_k^1)^{\otimes 4} \right\rangle\right], \tag{47}$$

where we use $\tilde{\theta}_k$ to indicate the point achieving the equality from the mean value theorem. Since $\mathbb{E}_\rho[\xi_k^1] = 0$, we have

$$\mathbb{E}_{\mu_k \otimes \rho}\left[\left\langle \nabla \frac{\delta J}{\delta \mu}[\mu_k, \mathcal{K}_*^+[\mu_k]], -\eta_1 g_k + \sqrt{2\eta_1 \tau}\xi_k^1 \right\rangle\right] = \mathbb{E}_{\mu_k}\left[\left\langle \nabla \frac{\delta J}{\delta \mu}[\mu_k, \mathcal{K}_*^+[\mu_k]], -\eta_1 g_k \right\rangle\right]. \tag{48}$$

Since $\|\nabla^2 \frac{\delta J}{\delta \mu}[\mu, \nu]\|_F < M_2$ for all $\mu, \nu$, we further have

$$\mathbb{E}_{\mu_k \otimes \rho}\left[\left\langle \nabla^2 \frac{\delta J}{\delta \mu}[\mu_k, \mathcal{K}_*^+[\mu_k]], (\eta_1 g_k + \sqrt{2\eta_1 \tau}\xi_k^1)^{\otimes 2} \right\rangle\right] \leq M_2 \eta_1^2 \mathbb{E}_{\mu_k}[\|g_k\|_2^2] + 2\eta_1 \tau \mathbb{E}_{\mu_k}\left[\mathrm{Tr}\left(\nabla^2 \frac{\delta J}{\delta \mu}[\mu_k, \nu_k]\right)\right]$$
$$= M_2 \eta_1^2 \mathbb{E}_{\mu_k}[\|g_k\|_2^2] - 2\eta_1 \tau \mathbb{E}_{\mu_k}[\langle g_k, \nabla \log \mu_k \rangle] + 2\eta_1 \tau^2 d, \tag{49}$$

where the equality follows from (24).

Similarly, we can derive an upper bound for the third term and the fourth term:

$$\mathbb{E}_{\mu_k \otimes \rho}\left[\left\langle \nabla^3 \frac{\delta J}{\delta \mu}[\mu_k, \mathcal{K}_*^+[\mu_k]], (-\eta_1 g_k + \sqrt{2\eta_1 \tau}\xi_k^1)^{\otimes 3}\right\rangle\right] \leq M_3(\eta_1^3 \mathbb{E}_{\mu_k}[\|g_k\|_2^3] + 6\eta_1^2 \tau d \mathbb{E}_{\mu_k}[\|g_k\|_2]).$$

and

$$\mathbb{E}_{\mu_k \otimes \rho}\left[\left\langle \nabla^4 \frac{\delta J}{\delta \mu}[\mu_k, \mathcal{K}_*^+[\mu_k]](\tilde{\theta}_k), (-\eta_1 g_k + \sqrt{2\eta_1 \tau}\xi_k^1)^{\otimes 4}\right\rangle\right]$$
$$\leq M_4(\eta_1^4 \mathbb{E}_{\mu_k}[\|g_k\|_2^4] + 4\sqrt{2}\eta_1^{3.5}\tau^{1/2}m_1\mathbb{E}_{\mu_k}[\|g_k\|_2^3] + 12d\eta_1^3\tau\mathbb{E}_{\mu_k}[\|g_k\|_2^2] + 8\sqrt{2}m_3\tau^{3/2}\eta_1^{5/2}\mathbb{E}_{\mu_k}[\|g_k\|_2] + 4\eta_1^2\tau^2 d(d+2)).$$

Where $m_1 = \sqrt{2}\frac{\Gamma((d+1)/2)}{\Gamma(d/2)}, m_3 = 2^{3/2}\frac{\Gamma((d+3)/2)}{\Gamma(d/2)}$.

The second term of (43) is indeed

$$\int_{\mathbb{R}^d} \frac{\delta^2 J}{\delta \mu^2}[\tilde{\mu}, \mathcal{K}_*^+[\mu_k]](\theta, \theta')\mathrm{d}(\mu_{k+1} - \mu_k)(\theta')\mathrm{d}(\mu_{k+1} - \mu_k)(\theta)$$
$$= \mathbb{E}_{\mu_{k+1}-\mu_k}\left[\int_{\mathbb{R}^d} \frac{\delta^2 J}{\delta \mu^2}[\tilde{\mu}, \mathcal{K}_*^+[\mu_k]](\theta, \theta')\mathrm{d}(\mu_{k+1} - \mu_k)(\theta')\right]$$
$$= \mathbb{E}_{\mu_{k+1}-\mu_k}\left[\mathbb{E}_{\mu_k \otimes \rho}\left[\nabla_{\theta'}\frac{\delta^2 J}{\delta \mu^2}[\tilde{\mu}, \mathcal{K}_*^+[\mu_k]](\theta, \tilde{\theta}')(\eta_1 g_k + \sqrt{2\eta_1\tau}\xi_k^1)\right]\right]$$
$$= \mathbb{E}_{\mu_{k+1}-\mu_k}\left[\mathbb{E}_{\mu_k \otimes \rho}\left[\nabla_{\theta'}\frac{\delta^2 J}{\delta \mu^2}[\tilde{\mu}, \mathcal{K}_*^+[\mu_k]](\theta, \tilde{\theta}')(\eta_1 g_k + \sqrt{2\eta_1\tau}\xi_k^1)\right]\right] \tag{50}$$
$$= \mathbb{E}_{\mu_{k+1}-\mu_k}\left[\mathbb{E}_{\mu_k}\left[\nabla_{\theta'}\frac{\delta^2 J}{\delta \mu^2}[\tilde{\mu}, \mathcal{K}_*^+[\mu_k]](\theta, \tilde{\theta}')\eta_1 g_k\right]\right]$$
$$= \mathbb{E}_{\mu_k}\left[\mathbb{E}_{\mu_k}\left[\nabla_\theta \nabla_{\theta'}^\top \frac{\delta^2 J}{\delta \mu^2}[\tilde{\mu}, \mathcal{K}_*^+[\mu_k]](\theta, \tilde{\theta}')\eta_1 g_k\right]\eta_1 g_k\right]$$
$$\leq C_1\eta_1^2(\mathbb{E}_{\mu_k}[\|g_k\|_2])^2$$
$$\leq C_1\eta_1^2\mathbb{E}_{\mu_k}[\|g_k\|_2^2].$$

Then we analyse the first term $E(\mu_{k+1}, \mathcal{K}_*^+[\mu_{k+1}] - E(\mu_{k+1}, \mathcal{K}_*^+[\mu_k])$, by concavity of $J(\mu, \cdot) + \tau\mathcal{H}(\cdot)$:

$$E(\mu_{k+1}, \mathcal{K}_*^+[\mu_{k+1}]) - E(\mu_{k+1}, \mathcal{K}_*^+[\mu_k])$$
$$\leq \int_{\mathbb{R}^d}\left(\frac{\delta J}{\delta \nu}[\mu_{k+1}, \mathcal{K}_*^+[\mu_k]] - \frac{\tau}{2}\|\omega\|_2^2 - \tau\log\mathcal{K}_*^+[\mu_k]\right)\mathrm{d}(\mathcal{K}_*^+[\mu_{k+1}] - \mathcal{K}_*^+[\mu_k])$$
$$= \int_{\mathbb{R}^d}\left(\frac{\delta J}{\delta \nu}[\mu_k, \mathcal{K}_*^+[\mu_k]] + \int_{\mathbb{R}^d}\frac{\delta^2 J}{\delta \nu \delta \mu}[\tilde{\mu}, \mathcal{K}_*^+[\mu_k]]\mathrm{d}(\mu_{k+1} - \mu_k) - \frac{\tau}{2}\|\omega\|_2^2 - \tau\log\mathcal{K}_*^+[\mu_k]\right)\mathrm{d}(\mathcal{K}_*^+[\mu_{k+1}] - \mathcal{K}_*^+[\mu_k])$$
$$= \int_{\mathbb{R}^d}\int_{\mathbb{R}^d}\frac{\delta^2 J}{\delta \mu \delta \nu}[\tilde{\mu}, \mathcal{K}_*^+[\mu_k]]\mathrm{d}(\mu_{k+1} - \mu_k)\mathrm{d}(\mathcal{K}_*^+[\mu_{k+1}] - \mathcal{K}_*^+[\mu_k])$$

$$\tag{51}$$

The last equality is since $\frac{\delta J}{\delta \nu}[\mu_k, \mathcal{K}_*^+[\mu_k]] - \frac{\tau}{2}\|\omega\|_2^2 - \tau\log\mathcal{K}_*^+[\mu_k] = \tau\log\mathcal{Z}^*[\mu_k]$, which is a constant relative to $\omega$.

Now

$$
\begin{aligned}
&\int_{\mathbb{R}^d}\int_{\mathbb{R}^d}\frac{\delta^2 J}{\delta\mu\delta\nu}[\tilde{\mu},\mathcal{K}_*^+[\mu_k]]\mathrm{d}(\mu_{k+1}-\mu_k)\mathrm{d}(\mathcal{K}_*^+[\mu_{k+1}]-\mathcal{K}_*^+[\mu_k])\\
&=\int_{\mathbb{R}^d}\mathbb{E}_{\mu_{k+1}-\mu_k}\left[\frac{\delta^2 J}{\delta\mu\delta\nu}[\tilde{\mu},\mathcal{K}_*^+[\mu_k]]\right]\mathrm{d}(\mathcal{K}_*^+[\mu_{k+1}]-\mathcal{K}_*^+[\mu_k])\\
&=\int_{\mathbb{R}^d}\mathbb{E}_{\mu_k}\left[\nabla_\theta\frac{\delta^2 J}{\delta\mu\delta\nu}[\tilde{\mu},\mathcal{K}_*^+[\mu_k]](\tilde{\theta})(\eta_1 g_k+\sqrt{2\eta_1\tau}\xi_k^1)\right]\mathrm{d}(\mathcal{K}_*^+[\mu_{k+1}]-\mathcal{K}_*^+[\mu_k])\\
&\leq\eta_1 C_2\mathbb{E}_{\mu_k}[\|g_k\|_2]\mathsf{TV}(\mathcal{K}_*^+[\mu_{k+1}],\mathcal{K}_*^+[\mu_k])\\
&\leq\sqrt{2}\eta_1 C_2\mathbb{E}_{\mu_k}[\|g_k\|_2]\sqrt{\mathsf{KL}(\mathcal{K}_*^+[\mu_{k+1}]|\mathcal{K}_*^+[\mu_k])}\\
&\leq\sqrt{2}\eta_1 C_2\mathbb{E}_{\mu_k}[\|g_k\|_2]\cdot\sqrt{\frac{1}{2\alpha\tau^2}\mathbb{E}_{\mathcal{K}_*^+[\mu_{k+1}]}\left[\left\|\int\nabla_\omega\frac{\delta^2 J}{\delta\mu\delta\nu}[\mu_k,\mathcal{K}_*^+[\tilde{\mu}]]\mathrm{d}(\mu_{k+1}-\mu_k)\right\|_2^2\right]}\\
&\leq\sqrt{2}\eta_1 C_2\mathbb{E}_{\mu_k}[\|g_k\|_2]\cdot\sqrt{\frac{1}{2\alpha\tau^2}C_1^2\eta_1^2(\mathbb{E}_{\mu_k}[\|g_k\|_2])^2}\\
&\leq\frac{\eta_1^2}{\alpha^{1/2}\tau}C_1 C_2\mathbb{E}_{\mu_k}[\|g_k\|_2^2].
\end{aligned}
\tag{52}
$$

Combining these inequalities with the result in Lemma 7, we finally get

$$
\begin{aligned}
&\mathcal{L}_1(\mu_{k+1})-\mathcal{L}_1(\mu_k)\\
&=-\eta_1\mathbb{E}_{\mu_k}[\langle f_k,g_k\rangle]-\eta_1\tau\mathbb{E}_{\mu_k}[\langle f_k+g_k,\nabla\log\mu_k\rangle]+\eta_1\tau^2\mathbb{E}_{\mu_k}[\|\nabla\log\mu_k\|_2^2]+\frac{\eta_1^2 C_1}{2}\mathbb{E}_{\mu_k}[\|g_k\|_2^2]+\Gamma_0\eta_1^2\\
&\leq-\frac{\eta_1}{2}\mathbb{E}_{\mu_k}[\|f_k+\nabla\log\mu_k\|_2^2]+\frac{\eta_1}{2}\mathbb{E}_{\mu_k}[\|g_k-f_k\|_2^2]+\Gamma_0\eta_1^2,
\end{aligned}
\tag{53}
$$

where

$$
\Gamma_0=(\frac{M_2+\tau}{2}\mathbb{E}_{\mu_k}[\|g_k\|_2^2]+\frac{M_3}{6}(\eta_1\mathbb{E}_{\mu_k}[\|g_k\|_2^3]+6\tau d(\mathbb{E}_{\mu_k}[\|g_k\|_2])+M_2^2\tau/2+2d^2 M_4\tau^2+\frac{C_1 C_2}{\alpha^{1/2}\tau}\mathbb{E}_{\mu_k}[\|g_k\|_2^2]
$$
$$
+M_4(\eta_1^4\mathbb{E}_{\mu_k}[\|g_k\|_2^4]+4\sqrt{2}\eta_1^{3.5}\tau^{1/2}m_1\mathbb{E}_{\mu_k}[\|g_k\|_2^3]+12d\eta_1^3\tau\mathbb{E}_{\mu_k}[\|g_k\|_2^2]+8\sqrt{2}m_3\tau^{3/2}\eta_1^{5/2}\mathbb{E}_{\mu_k}[\|g_k\|_2]+4\eta_1^2\tau^2 d(d+2)).
$$

The last inequality is since $\eta_1<\frac{1}{C_1}$. Therefore, we complete the proof. $\qquad\square$

*Proof of Lemma 1.* By the definition of $E$, we note that

$$
\begin{aligned}
E(\mu_{k+1},\nu_{k+1})-E(\mu_{k+1},\nu_k)&=(J(\mu_{k+1},\nu_{k+1})-J(\mu_{k+1},\nu_k))-\tau(\mathsf{KL}(\nu_{k+1}|\rho^\nu)-\mathsf{KL}(\nu_k|\rho^\nu))\\
&=(J(\mu_{k+1},\nu_{k+1})-J(\mu_{k+1},\nu_k))+\tau(\mathcal{H}(\nu_{k+1})-\mathcal{H}(\nu_k))+\frac{\tau}{2}(\mathbb{E}_{\nu_{k+1}}[\|\omega_{k+1}\|_2^2]-\mathbb{E}_{\nu_k}[\|\omega_k\|_2^2]).
\end{aligned}
$$

Note that the second term $\mathcal{H}(\nu_{k+1})-\mathcal{H}(\nu_k)$ can be bounded via Lemma 7. In the following, we focus on term $\frac{\tau}{2}\mathbb{E}_{\nu_{k+1}-\nu_k}[\|\omega_k\|_2^2]$, Let $\rho=\mathcal{N}(0,I_d)$ be a standard multivariate Gaussian distribution, then:

$$
\begin{aligned}
\frac{\tau}{2}\mathbb{E}_{\nu_{k+1}-\nu_k}[\|\omega\|_2^2]&=\frac{\tau}{2}\mathbb{E}_{\nu_k\otimes\rho}[\|\omega_k+\eta_2 h_k+\sqrt{2\eta_2\tau}\xi_k^2\|_2^2-\|\omega_k\|_2^2]\\
&=\frac{\tau}{2}\mathbb{E}_{\nu_k}[2\eta_2\langle\omega_k,h_k\rangle+\eta_2^2\|h_k\|_2^2]+\eta_2\tau^2 d.
\end{aligned}
\tag{54}
$$

By second order expansion with functional $J(\mu_{k+1},\cdot)$, it holds that

$$
J(\mu_{k+1},\nu_{k+1})-J(\mu_{k+1},\nu_k)=\mathbb{E}_{\nu_{k+1}-\nu_k}\left[\frac{\delta J}{\delta\nu}[\mu_{k+1},\nu_k](\omega_k)\right]+\frac{1}{2}\int_{\mathbb{R}^{d2}}\frac{\delta^2 J}{\delta\nu^2}[\mu_{k+1},\tilde{\nu}](\omega,\omega')\mathrm{d}(\nu_{k+1}-\nu_k)(\omega)\mathrm{d}(\nu_{k+1}-\nu_k)(\omega')
\tag{55}
$$

Define

$$
\frac{\tilde{\delta J}}{\delta\nu}[\mu,\nu](\omega)=\mathbb{E}_{\xi\sim\rho}\left[\frac{\delta J}{\delta\nu}[\mu,\nu](\omega+\sqrt{2\eta_2\tau}\xi)\right].
\tag{56}
$$

By Taylor expansion with Lagrangian remainder, the first part on the right-hand side of (55) becomes

$$\mathbb{E}_{\nu_{k+1}-\nu_k}\left[\frac{\delta J}{\delta \nu}[\mu_{k+1},\nu_k]\right] \tag{57}$$

$$=\mathbb{E}_{\bar{\nu}_{k+1}}\left[\frac{\tilde{\delta J}}{\delta \nu}[\mu_{k+1},\nu_k](\bar{\omega}_{k+1})\right] - \mathbb{E}_{\nu_k}\left[\frac{\delta J}{\delta \nu}[\mu_{k+1},\nu_k](\omega_k)\right]$$

$$=\mathbb{E}_{\nu_k}\left[\frac{\tilde{\delta J}}{\delta \nu}[\mu_{k+1},\nu_k](\omega_k)\right] - \mathbb{E}_{\nu_k}\left[\frac{\delta J}{\delta \nu}[\mu_{k+1},\nu_k](\omega_k)\right] \tag{58}$$

$$=\mathbb{E}_{\nu_k\otimes\rho}\left[\left\langle\nabla\frac{\delta J}{\delta \nu}[\mu_{k+1},\nu_k],\eta_2 h_k + \sqrt{2\eta_2\tau}\xi_k^2\right\rangle + \frac{1}{2}\left\langle\nabla^2\frac{\delta J}{\delta \nu}[\mu_{k+1},\nu_k],(\eta_2 h_k + \sqrt{2\eta_2\tau}\xi_k^2)^{\otimes 2}\right\rangle\right] \tag{59}$$

$$+\frac{1}{6}\mathbb{E}_{\nu_k\otimes\rho}\left[\left\langle\nabla^3\frac{\delta J}{\delta \nu}[\nu_{k+1},\nu_k],(\eta_2 h_k + \sqrt{2\eta_2\tau}\xi_k^2)^{\otimes 3}\right\rangle\right], \tag{60}$$

$$+\frac{1}{24}\mathbb{E}_{\nu_k\otimes\rho}\left[\left\langle\nabla^4\frac{\delta J}{\delta \nu}[\mu_{k+1},\nu_k](\tilde{\omega}_k),(\eta_2 h_k + \sqrt{2\eta_2\tau}\xi_k^2)^{\otimes 4}\right\rangle\right], \tag{61}$$

where we use $\tilde{\omega}_k$ to indicate the point achieving the equality from the mean value theorem. Since $\mathbb{E}_\rho[\xi_k^2]=0$, we have

$$\mathbb{E}_{\nu_k\otimes\rho}\left[\left\langle\nabla\frac{\delta J}{\delta \nu}[\mu_{k+1},\nu_k],\eta_2 h_k + \sqrt{2\eta_2\tau}\xi_k^2\right\rangle\right] = \mathbb{E}_{\nu_k\otimes\rho}\left[\left\langle\nabla\frac{\delta J}{\delta \nu}[\mu_{k+1},\nu_k],\eta_2 h_k\right\rangle\right]. \tag{62}$$

Since $\|\nabla^2\frac{\delta J}{\delta \nu}[\mu,\nu]\|_F < M_2$, we further have

$$\mathbb{E}_{\nu_k\otimes\rho}\left[\left\langle\nabla^2\frac{\delta J}{\delta \nu}[\mu_{k+1},\nu_k],(\eta_2 h_k + \sqrt{2\eta_2\tau}\xi_k^2)^{\otimes 2}\right\rangle\right] \geq -M_2\eta_2^2\mathbb{E}_{\nu_k}[\|h_k\|_2^2] + 2\eta_2\tau\mathbb{E}_{\nu_k}\left[\mathrm{Tr}\left(\nabla^2\frac{\delta J}{\delta \nu}[\mu_{k+1},\nu_k]\right)\right]$$
$$= -M_2\eta_2^2\mathbb{E}_{\nu_k}[\|h_k\|_2^2] - 2\eta_2\tau\mathbb{E}_{\nu_k}[\langle h_k,\nabla\log\nu_k\rangle] - 2\eta_2\tau^2 d, \tag{63}$$

where the equality follows from (24).

Similarly, we can derive an upper bound for the third term:

$$\mathbb{E}_{\nu_k\otimes\rho}\left[\left\langle\nabla^3\frac{\delta J}{\delta \nu}[\mu_{k+1},\nu_k],(\eta_2 h_k + \sqrt{2\eta_2\tau}\xi_k^2)^{\otimes 3}\right\rangle\right] \geq -M_3(\eta_2^3\mathbb{E}_{\nu_k}[\|h_k\|_2^3] + 6\eta_2^2\tau d\mathbb{E}_{\nu_k}[\|h_k\|_2]).$$

Plugging in the above inequalities to (57), we arrive at

$$\mathbb{E}_{\nu_{k+1}-\nu_k}\left[\frac{\delta J}{\delta \nu}[\mu_{k+1},\nu_k]\right] \geq \eta_2\mathbb{E}_{\nu_k}[\|h_k\|_2^2] - \eta_2\tau\mathbb{E}_{\nu_k}[\langle h_k,\nabla\log\nu_k\rangle]$$
$$- \eta_2^2(\frac{M_2}{2}\mathbb{E}_{\nu_k}[\|h_k\|_2^2] + \frac{M_3}{6}(\eta_2\mathbb{E}_{\nu_k}[\|h_k\|_2^3] + 6\tau d\mathbb{E}_{\nu_k}[\|h_k\|_2]).$$

The second term is indeed

$$\frac{1}{2}\int_{\mathbb{R}^d}\frac{\delta^2 J}{\delta\nu^2}[\mu_{k+1},\tilde{\nu}](\omega,\omega')\mathrm{d}(\nu_{k+1}-\nu_k)\mathrm{d}(\nu_{k+1}-\nu_k) \geq -\frac{C_1\eta_2^2}{2}\mathbb{E}_{\nu_k}[\|h_k\|_2^2] \tag{64}$$

which is similar to (50). Combining with the bound of $\tau(\mathcal{H}(\nu_{k+1}) - \mathcal{H}(\nu_k))$ as established in Lemma 7, we then obtain

$$E(\mu_k,\nu_{k+1}) - E(\mu_k,\nu_k) \geq \eta_2\mathbb{E}_{\nu_k}[\|h_k\|_2^2] - 2\eta_2\tau\mathbb{E}_{\nu_k}[\langle h_k,\nabla\log\nu_k\rangle] + \eta_2\tau^2\mathbb{E}_{\nu_k}[\|\nabla\log\nu_k\|_2^2] - \Gamma_2\eta_2^2$$
$$= \eta_2\mathbb{E}_{\nu_k}[\|h_k - \tau\nabla\log\nu_k\|_2^2] - \Gamma_2\eta_2^2,$$

where

$$\Gamma_2 = (\frac{M_2+\tau}{2}\mathbb{E}_{\nu_k}[\|h_k\|_2^2] + \frac{M_3}{6}(\eta_2\mathbb{E}_{\nu_k}[\|h_k\|_2^3] + 6\tau d(\mathbb{E}_{\nu_k}[\|h_k\|_2^2] + 1))) + (M_2+\tau)^2\tau/2 + 2d^2 M_4\tau^2 + \frac{C_1}{2}\mathbb{E}_{\nu_k}[\|h_k\|_2^2]$$
$$+ M_4(\eta_2^4\mathbb{E}_{\nu_k}[\|h_k\|_2^4] + 4\sqrt{2}\eta_2^{3.5}\tau^{1/2}m_1\mathbb{E}_{\nu_k}[\|h_k\|_2^3] + 12d\eta_2^3\tau\mathbb{E}_{\nu_k}[\|h_k\|_2^2] + 8\sqrt{2}m_3\tau^{3/2}\eta_2^{5/2}\mathbb{E}_{\nu_k}[\|h_k\|_2] + 4\eta_2^2\tau^2 d(d+2)). \tag{65}$$

where $\mathbb{E}_{\nu_k}[\|h_k\|_2^i], i = 1, 2, 3, 4$ terms can be further bounded by proposition 10. This gives rise to the desired result. $\quad\square$

*Proof of Lemma 2.* By definition of $\mathcal{K}_\mu^+[\nu]$, we have

$$\mathbb{E}_{\nu_k}[\|h_k - \tau\nabla\log\nu_k\|_2^2] = \mathbb{E}_{\nu_k}[\|\tau\nabla\log\mathcal{K}_{\mu_{k+1}}^+[\nu_k] - \tau\nabla\log\nu_k\|_2^2] = \tau^2 I(\nu_k|\mathcal{K}_{\mu_{k+1}}^+[\nu_k]|).$$

Applying the log-Sobolev inequality for $\mathcal{K}_{\mu_{k+1}}^+[\nu_k]$ as well as Lemma 8, we then obtain

$$\tau^2 I(\nu_k|\mathcal{K}_{\mu_{k+1}}^+[\nu_k]|) \geq 2\alpha\tau^2\mathsf{KL}(\nu_k|\mathcal{K}_{\mu_{k+1}}^+[\nu_k]) \geq 2\alpha\tau\mathcal{L}_2(\mu_{k+1},\nu_k),$$

which implies that $\mathbb{E}_{\nu_k}[\|h_k - \tau\nabla\log\nu_k\|_2^2] \geq 2\alpha\tau\mathcal{L}_2(\mu_{k+1},\nu_k)$.

Based on Lemma 1,

$$\begin{aligned}
&\mathcal{L}_2(\mu_{k+1},\nu_{k+1})\\
&= E^*(\mu_{k+1}) - E(\mu_{k+1},\nu_{k+1})\\
&= E^*(\mu_{k+1}) - E(\mu_{k+1},\nu_k) + E(\mu_{k+1},\nu_k) - E(\mu_{k+1},\nu_{k+1})\\
&\leq E^*(\mu_{k+1}) - 2\eta_2\tau\alpha\mathcal{L}_2(\mu_{k+1},\nu_k) - E(\mu_{k+1},\nu_k) + \Gamma_2\eta_2^2\\
&= (1 - 2\eta_2\tau\alpha)\mathcal{L}_2(\mu_{k+1},\nu_k) + \Gamma_2\eta_2^2.
\end{aligned} \tag{66}$$

Hence get the desired result. $\qquad\square$

*Proof of Proposition 2.* Lemma 2 shows that

$$\begin{aligned}
\mathcal{L}_2(\mu_{k+1},\nu_{k+1}) &\leq (1 - 2\eta_2\tau\alpha)\mathcal{L}_2(\mu_{k+1},\nu_k) + \Gamma_2\eta_2^2\\
&= (1 - 2\eta_2\tau\alpha)(\mathcal{L}_2(\mu_k,\nu_k) + E(\mu_k,\nu_k) - E(\mu_{k+1},\nu_k) + E^*(\mu_{k+1}) - E^*(\mu_k)) + \Gamma_2\eta_2^2.
\end{aligned} \tag{67}$$

By Lemma 1,

$$E(\mu_k,\nu_k) - E(\mu_{k+1},\nu_k) \leq \eta_1\mathbb{E}_{\mu_k}[\|g_k + \nabla\log\mu_k\|_2^2] + \Gamma_1\eta_1^2. \tag{68}$$

By Proposition 1, we further have

$$E^*(\mu_{k+1}) - E^*(\mu_k) \leq -\frac{\eta_1}{2}\mathbb{E}_{\mu_k}[\|f_k + \nabla\log\mu_k\|_2^2] + \frac{\eta_1}{2}\mathbb{E}_{\mu_k}[\|g_k - f_k\|_2^2] + \Gamma_0\eta_1^2.$$

Combining the above inequalities, inequality (67) then becomes

$$\begin{aligned}
\mathcal{L}_2(\mu_{k+1},\nu_{k+1}) &\leq (1 - 2\eta_2\tau\alpha)\mathcal{L}_2(\mu_k,\nu_k) + (1 - 2\eta_2\tau\alpha)\eta_1\mathbb{E}_{\mu_k}[\|g_k + \nabla\log\mu_k\|_2^2]\\
&\quad - (1 - 2\eta_2\tau\alpha)\frac{\eta_1}{2}\mathbb{E}_{\mu_k}[\|f_k + \nabla\log\mu_k\|_2^2] + (1 - 2\eta_2\tau\alpha)\frac{\eta_1}{2}\mathbb{E}_{\mu_k}[\|g_k - f_k\|_2^2]\\
&\quad + \Gamma_2\eta_2^2 + (1 - 2\eta_2\tau\alpha)(\Gamma_1 + \Gamma_0)\eta_1^2,
\end{aligned} \tag{69}$$

which completes the proof. $\qquad\square$

*Proof of Theorem 1.* Combining (53) and (69), we have

$$\begin{aligned}
\mathcal{L}_1(\mu_{k+1}) + \lambda\mathcal{L}_2(\mu_{k+1},\nu_{k+1}) &\leq \mathcal{L}_1(\mu_k) - \frac{\eta_1}{2}(1 + \lambda(1 - \eta_2\tau\alpha))\mathbb{E}_{\mu_k}[\|f_k + \nabla\log\mu_k\|_2^2] + \lambda(1 - \eta_2\tau\alpha)\mathcal{L}_2(\mu_k,\nu_k)\\
&\quad + \lambda(1 - \eta_2\tau\alpha)\eta_1\mathbb{E}_{\mu_k}[\|g_k + \nabla\log\mu_k\|_2^2] + (\lambda(1 - \eta_2\tau\alpha) + 1)\frac{\eta_1}{2}\mathbb{E}_{\mu_k}[\|g_k - f_k\|_2^2]\\
&\quad + \lambda(\Gamma_2\eta_2^2 + (1 - 2\eta_2\tau\alpha)(\Gamma_1 + \Gamma_0)\eta_1^2) + \Gamma_1\eta_1^2.
\end{aligned} \tag{70}$$

According to the Pinsker's inequality, it holds that $2\mathsf{KL}(\mu|\nu) \geq \mathrm{TV}^2(\mu,\nu)$ where $\mathrm{TV}(\mu,\nu) = \|\mu-\nu\|_1$ is the total-variation distance between $\mu$ and $\nu$. Since $J$ has bounded second-order Wasserstein gradients, we have

$$\mathbb{E}_{\mu_k}[\|g_k - f_k\|_2^2] = \int_\Theta \left|\nabla\frac{\delta J}{\delta\mu}[\mu_k,\mathcal{K}_*^+[\mu_k]] - \nabla\frac{\delta J}{\delta\mu}[\mu_k,\nu_k]\right|^2 \mathrm{d}\mu_k \leq C_1^2\mathrm{TV}^2(\mathcal{K}_*^+[\mu_k],\nu_k) \leq 2C_1^2\mathsf{KL}(\nu_k|\mathcal{K}_*^+[\mu_k]), \tag{71}$$

which, by invoking Lemma 8, further leads to

$$\mathbb{E}_{\mu_k}[\|g_k - f_k\|_2^2] \leq 2C_1^2\mathsf{KL}(\nu_k|\mathcal{K}_*^+[\mu_k]) \leq \frac{2C_1^2}{\tau}\mathcal{L}_2(\mu_k,\nu_k). \tag{72}$$

By Assumption 3, $\mathcal{K}_\mu^-[\nu]$ satisfies the log-Sobolev inequality with parameter $\alpha$, and hence,

$$\mathbb{E}_{\mu_k}[\|f_k + \nabla \log \mu_k\|_2^2] = \tau^2 I(\mu_k | \mathcal{K}_{\mu_k}^-[\mathcal{K}_*^+[\mu_k]]) \geq 2\alpha\tau^2 \mathsf{KL}(\mu_k | \mathcal{K}_{\mu_k}^-[\mathcal{K}_*^+[\mu_k]]) \geq 2\alpha\tau \mathcal{L}_1(\mu_k),$$

where the last inequality follows from Lemma 8. Additionally, applying the Young's inequality, we obtain

$$\mathbb{E}_{\mu_k}[\|g_k + \nabla \log \mu_k\|_2^2] \leq 2(\mathbb{E}_{\mu_k}[\|f_k + \nabla \log \mu_k\|_2^2 + \mathbb{E}_{\mu_k}[\|g_k - f_k\|_2^2]). \tag{73}$$

Combining the above inequalities then yields

$$
\begin{aligned}
\mathcal{L}_1(\mu_{k+1}) + \lambda\mathcal{L}_2(\mu_{k+1}, \nu_{k+1}) &\leq \left(1 - \left(\frac{\eta_1}{2}(1 + \lambda(1 - 2\eta_2\tau\alpha)) + 2\eta_1\lambda(1 - 2\eta_2\tau\alpha)\right)2\alpha\tau\right)\mathcal{L}_1(\mu_k) \\
&+ \left(\lambda(1 - 2\eta_2\tau\alpha) + \left((\lambda(1 - 2\eta_2\tau\alpha) + 1)\frac{\eta_1}{2} + 2\lambda(1 - 2\eta_2\tau\alpha)\eta_1\right)\frac{2C_1^2}{\tau}\right)\mathcal{L}_2(\mu_k, \nu_k) \\
&+ \lambda(\Gamma_2\eta_2^2 + (1 - 2\eta_2\tau\alpha)(\Gamma_1 + \Gamma_0)\eta_1^2) + \Gamma_1\eta_1^2.
\end{aligned} \tag{74}
$$

By noting that

$$1 - \left(\frac{\eta_1}{2}(1 + \lambda(1 - 2\eta_2\tau\alpha)) + 2\eta_1\lambda(1 - 2\eta_2\tau\alpha)\right)2\alpha\tau \leq 1 - \eta_1\alpha\tau,$$

since $1 - 2\eta_2\tau\alpha > 0$, as well as

$$
\begin{aligned}
&\lambda(1 - 2\eta_2\tau\alpha) + \left((\lambda(1 - 2\eta_2\tau\alpha) + 1)\frac{\eta_1}{2} + 2\lambda(1 - 2\eta_2\tau\alpha)\eta_1\right)\frac{2C_1^2}{\tau} \\
&\leq \lambda(1 - 2\eta_2\tau\alpha + \frac{\eta_1}{2}(1 - 2\eta_2\tau\alpha) + \frac{\eta_1}{2\lambda} + 2\eta_1(1 - 2\eta_2\tau\alpha)) \\
&\leq \lambda(1 - \eta_1\alpha\tau),
\end{aligned}
$$

we can rewrite inequality (74) into

$$\mathcal{L}_1(\mu_{k+1}) + \lambda\mathcal{L}_2(\mu_{k+1}, \nu_{k+1}) \leq (1 - \eta_1\tau\alpha)(\mathcal{L}_1(\mu_k) + \lambda\mathcal{L}_2(\mu_k, \nu_k)) + \lambda(\Gamma_1\eta_2^2 + (1 - 2\eta_2\tau\alpha)(2\Gamma_2 + M_2 M_1^2)\eta_1^2) + \Gamma_2\eta_1^2.$$

By recursively applying the above inequality, we then obtain

$$\mathcal{L}(\mu_k) \leq (1 - \eta_1\tau\alpha)^k \mathcal{L}(\mu_0) + O(\eta_1),$$

where $O(\eta_1) = \frac{\lambda(\Gamma_1\eta_2^2 + (1 - 2\eta_2\tau\alpha)(2\Gamma_2 + M_2 M_1^2)\eta_1^2) + \Gamma_2\eta_1^2}{\eta_1\tau\alpha}$. This completes the proof. $\qquad\square$

*Proof of Corollary 1.* We proceed by bounding $\mathsf{KL}(\mu_k | \mu^*)$ and $\mathsf{KL}(\nu_k | \nu^*)$ separately. Lemma 8 and Theorem 1 allow us to bound $\mathsf{KL}(\mu_k | \mu^*)$ by

$$\tau\mathsf{KL}(\mu_k | \mu^*) \leq \mathcal{L}_1(\mu_k) \leq \mathcal{L}(\mu_k, \nu_k) \leq (1 - \eta_1\tau\alpha)^k \mathcal{L}(\mu_0, \nu_0) + R_1(\eta_1). \tag{75}$$

In the following, we focus on $\mathsf{KL}(\nu_k | \nu^*)$. By definition, it holds that

$$
\begin{aligned}
\tau\mathsf{KL}(\nu_k | \nu^*) &= \tau\mathsf{KL}(\nu_k | \mathcal{K}_*^+[\mu_k]) + \tau\int_\Omega (\log(\mathcal{K}_*^+[\mu_k]) - \log\nu^*)\mathrm{d}\nu_k \\
&= \tau\mathsf{KL}(\nu_k | \mathcal{K}_*^+[\mu_k]) + \tau\int_\Omega (\log\mathcal{K}_*^+[\mu_k] - \log\nu^*)\mathrm{d}(\nu_k - \nu^*) - \tau\mathcal{K}(\nu^* | \mathcal{K}_*^+[\nu_k]) \\
&\leq \tau\mathsf{KL}(\nu_k | \mathcal{K}_*^+[\mu_k]) + \tau\int_\Omega (\log\mathcal{K}_*^+[\mu_k] - \log\nu^*)\mathrm{d}(\nu_k - \nu^*).
\end{aligned} \tag{76}
$$

By noting that $\nu^* = \mathcal{K}_*^+[\mu^*]$, we then have

$$\log\mathcal{K}_*^+[\mu_k] - \log(\nu^*) = \tau^{-1}\left(\frac{\delta J}{\delta\nu}[\mu, \mathcal{K}_*^+[\mu]] - \frac{\delta J}{\delta\nu}[\mu^*, \mathcal{K}_*^+(\mu^*)]\right) - (\log\mathcal{Z}^*[\mu_k] - \log\mathcal{Z}^*[\mu^*]). \tag{77}$$

And the constant part $\log \mathcal{Z}^*[\mu_k] - \log \mathcal{Z}^*[\mu^*]$ equals to zero when integrate over $\nu_k - \nu^*$. This leads to

$$
\begin{aligned}
\tau \mathsf{KL}(\nu_k|\nu^*) &\leq \tau \mathsf{KL}(\nu_k|\mathcal{K}_*^+[\mu_k]) + \int_\Omega \int_\Theta \left\| \frac{\delta^2 J}{\delta\mu\delta\nu}(\mu,\nu) \right\|_\infty \mathrm{d}(\mu_k - \mu^*)\mathrm{d}(\nu_k - \nu^*) \\
&\leq \tau \mathsf{KL}(\nu_k|\mathcal{K}_*^+[\mu_k]) + C_0 \cdot \mathrm{TV}(\mu_k, \mu^*) \cdot \mathrm{TV}(\nu_k, \nu^*) \\
&\leq \tau \mathsf{KL}(\nu_k|\mathcal{K}_*^+[\mu_k]) + 2C_0 \cdot \sqrt{\mathsf{KL}(\mu_k|\mu^*)} \cdot \sqrt{\mathsf{KL}(\nu_k|\nu^*)} \\
&\leq \tau \mathsf{KL}(\nu_k|\mathcal{K}_*^+[\mu_k]) + \frac{\tau}{2}\mathsf{KL}(\nu_k|\nu^*) + \frac{2C_0^2}{\tau}\mathsf{KL}(\mu_k|\mu^*),
\end{aligned}
\tag{78}
$$

where the second inequality used Pinsker's inequality and the last inequality used Young's inequality. Combining the above inequalities we then arrive at

$$
\tau \mathsf{KL}(\nu_k|\nu^*) \leq 2\tau \mathsf{KL}(\nu_k|\mathcal{K}_*^+[\mu_k]) + \frac{4C_0^2}{\tau}\mathsf{KL}(\mu_k|\mu^*) \leq \left( \frac{2}{\lambda} + \frac{4C_0^2}{\tau^2} \right) \left( (1 - \eta_1\tau\alpha)^k \mathcal{L}(\mu_0, \nu_0) + R_1(\eta_1) \right),
\tag{79}
$$

which yields the desired result. Finally, by Otto & Villani (2000), every $\nu$ satisfies LSI with parameter $\alpha$ also satisfies Talagrand's inequality with parameter $\alpha$, where

$$
\mathsf{W}_2^2(\mu,\nu) \leq \frac{2}{\alpha}\mathsf{KL}(\mu|\nu).
$$

For any $\mu$. $\qquad\qquad\qquad\qquad\qquad\qquad\qquad\qquad\qquad\qquad\qquad\qquad\qquad\qquad\qquad\qquad\qquad\qquad\qquad\qquad\square$

*Proof of Theorem 2.* The proof mainly relies on a distributional update, and we take the update of $\mu$ as an example:

$$
\bar{\mu}_{k+1} = ((I_d - \eta_1 \hat{g}_k(\cdot, z_k))_\sharp \mu_k), \quad \mu_{k+1} = \bar{\mu}_{k+1} * \rho_k,
\tag{80}
$$

where $\hat{g}_k(\cdot, z_k)$ is the inexact gradient oracle generated by label $z_k$, and $\rho_k$ represents an independent $N(0, 2\eta_1\tau I)$.

Let $\mathcal{H}_k$ be the $\sigma$-algebra generated by $\{z_j\}_{j=1}^k$, then conditional on $\mathcal{H}_k$, we have

$$
\begin{aligned}
&\mathbb{E}[E(\mu_{k+1}, \nu_k)|\mathcal{H}_k] - E(\mu_k, \nu_k) \\
&= \left( \mathbb{E}\left[ J(\mu_{k+1}, \nu_k) + \frac{\tau}{2}\mathbb{E}_{\mu_{k+1}}[\|\theta\|_2^2] \Big| \mathcal{H}_k \right] - J(\mu_k, \nu_k) - \frac{\tau}{2}\mathbb{E}_{\mu_k}[\|\theta\|_2^2]) \right) - \tau(\mathbb{E}[\mathcal{H}(\mu_{k+1})|\mathcal{H}_k] - \mathcal{H}(\mu_k)).
\end{aligned}
\tag{81}
$$

We can analyze the difference term conditioned on $\mathcal{H}_k$ similar with exact gradient as Proposition 1,2 and Lemma 1,

Among them, every expectation term of $\hat{H}_k(\cdot, z_k)$ would be $\mathbb{E}[\mathbb{E}_{\nu_k}[\hat{h}_k(\cdot, z_k)]|\mathcal{H}_k] = h_k$ by the law of iterated expectations. For the fourth order moments, we first bound

$$
\begin{aligned}
\mathbb{E}_{\nu_k}[\|\hat{h}_k(\cdot, z_k) + \tau\omega_k\|_4^2] &\leq 8\mathbb{E}_{\nu_k}[\|\hat{h}_k(\cdot, z_k) - h_k\|_2^4] + 8\mathbb{E}_{\nu_k}[\|h_k + \tau\omega_k\|_2^4] \\
&= 8\zeta + 8M_1^4.
\end{aligned}
\tag{82}
$$

Then similar to Lemma 10, denoted the distribution of random variable $z_k$ be $\zeta_k$:

Then

$$
\begin{aligned}
\mathbb{E}_{\nu_k}[\|\omega_k + \eta_2\hat{h}_k(\cdot, z_k)\|_2^4] &\leq \mathbb{E}_{\nu_k}[((1 - 3/2\eta_2\tau + \eta_2^2\tau^2)\|\omega_k\|_2^2 + \eta_2^2(M_1^2 + \zeta^{1/2}))^2] \\
&= (1 - 3/2\eta_2\tau + \eta_2^2\tau^2)^2 \mathbb{E}_{\nu_k}[\|\omega_k\|_2^4] + 2\eta_2^4(M_1^2 + \zeta^{1/2})(1 - 3/2\eta_2\tau + \eta_2^2\tau^2)\mathbb{E}_{\nu_k}[\|\omega_k\|_2^2] + \eta_2^4(M_1^2 + \zeta^{1/2})^2.
\end{aligned}
$$

Then combining them together,

$$
\begin{aligned}
&\mathbb{E}_{\nu_k}[\|\omega_k\|_2^4] \\
&\leq (1 - 3/2\eta_2\tau + \eta_2^2\tau^2)^2 \mathbb{E}_{\nu_k}[\|\omega_k\|_2^4] + \eta_2(1 - 3/2\eta_2\tau + \eta_2^2\tau^2)(8\tau + 4\tau d + 2\eta_2^3(M_1^2 + \zeta^{1/2}))\mathbb{E}_{\nu_k}[\|\omega_k\|_2^2] \\
&\quad + \eta_2^4(M_1^2 + \zeta^{1/2})^2 + (8 + 4d)\eta_2^3\tau(M_1^2 + \zeta^{1/2}) + 16\eta_2^2\tau^2(d^2 + 2d) \\
&\leq (1 - \eta_2\tau)\mathbb{E}_{\nu_k}[\|\omega_k\|_2^4] \\
&\quad + \eta_2(2(4\tau + 2\tau d + \eta_2^3(M_1^2 + \zeta^{1/2}))^2/\tau + \eta_2^2(M_1^2 + \zeta^{1/2})^2 + (8 + 4d)\eta_2\tau(M_1^2 + \zeta^{1/2}) + 16\tau^2(d^2 + 2d))
\end{aligned}
\tag{83}
$$

Hence, by iteratively applying this inequality we can get

$$\mathbb{E}_{\nu_k}[\|\omega_k\|_2^4] \le \mathbb{E}_{\nu_k}[\|\omega_0\|_2^4] + \frac{1}{\eta_2\tau}\eta_2(2(4\tau + 2\tau d + \eta_2^3(M_1^2 + \zeta^{1/2}))^2/\tau + \eta_2^2(M_1^2 + \zeta^{1/2})^2 + (8 + 4d)\eta_2\tau(M_1^2 + \zeta^{1/2}) + 16\tau^2(d^2 + 2d)).$$

Then by $(a + b)^4 \le 8(a^4 + b^4)$ we can obtain the upper bound of fourth moment of gradient $h_k$:

$$\mathbb{E}_{\nu_k}[\|\hat{h}_k(\cdot, z_k)\|_2^4] \le 8(\mathbb{E}_{\nu_k}[\|\hat{h}_k(\cdot, z_k) + \tau\omega_k\|_2^4 + \|\tau\omega_k\|_2^4])$$
$$\le 8(M_1^4 + \zeta) + 8\tau^3(\mathbb{E}_{\nu_k}[\|\omega_0\|_2^4]$$
$$+ (2(4\tau + 2\tau d + \eta_2^3(M_1^2 + \zeta^{1/2}))^2/\tau + \eta_2^2(M_1^2 + \zeta^{1/2})^2 + (8 + 4d)\eta_2\tau(M_1^2 + \zeta^{1/2}) + 16\tau^2(d^2 + 2d))).$$

Denote $R_{h4} = 8(M_1^4 + \zeta) + 8\tau^3((2(4\tau + 2\tau d + \eta_2^3(M_1^2 + \zeta^{1/2}))^2/\tau + \eta_2^2(M_1^2 + \zeta^{1/2}) + (8 + 4d)\eta_2\tau(M_1^2 + \zeta^{1/2}) + 16\tau^2(d^2 + 2d)))$.

For $\mathbb{E}_{\mu_k}[\|\hat{g}_k(\cdot, z_k)\|_2^4]$, we can similarly obtain

$$\mathbb{E}_{\mu_k}[\|\hat{g}_k(\cdot, z_k)\|_2^4] \le \mathbb{E}_{\mu_0}[\|\theta_0\|_2^4] + R_{g4},$$

where $R_{g4} = 8(M_1^4 + \zeta) + 8\tau^3((2(4\tau + 2\tau d + \eta_1^3(M_1^2 + \zeta^{1/2}))^2/\tau + \eta_1^2(M_1^2 + \zeta^{1/2}) + (8 + 4d)\eta_1\tau(M_1^2 + \zeta^{1/2}) + 16\tau^2(d^2 + 2d)))$.

Hence, by repeating the proof process of MFL-DA with exact gradients, we can get a result similar to exact gradient one, we give a track to whole proof process for 1, for first order term as (48), (62),

$$\mathbb{E}\left[\mathbb{E}_{\mu_k}\left[\left\langle \nabla\frac{\delta J}{\delta\mu}[\mu_k, \mathcal{K}_*^+[\mu_k]](\theta), -\eta_1\hat{g}_k(\theta, z_k)\right\rangle\right]\bigg|\mathscr{H}_k\right] = \mathbb{E}_{\mu_k}\left[\left\langle \nabla\frac{\delta J}{\delta\mu}[\mu_k, \mathcal{K}_*^+[\mu_k]], -\eta_1 g_k\right\rangle\right].$$

Since $\hat{g}_k, \hat{h}_k$ are unbiased estimator, hence introduce no error term.

For weight decay terms (42),(54), we have

$$\mathbb{E}\left[\frac{\tau}{2}\mathbb{E}_{\mu_{k+1} - \mu_k}[\|\theta\|_2^2]\bigg|\mathscr{H}_k\right] = \mathbb{E}\left[\frac{\tau}{2}\mathbb{E}_{\mu_k}[2\eta_1\langle\theta_k, \hat{g}_k(\theta_k; z_k)\rangle + \eta_1^2\|\hat{g}_k(\theta_k; z_k)\|_2^2] + \eta_1\tau^2 d\bigg|\mathscr{H}_k\right]$$
$$\le \eta_1\tau\mathbb{E}_{\mu_k}[\langle\theta_k, g_k\rangle] + \eta_1\tau^2 d + \frac{\tau}{2}\eta_1^2\mathbb{E}_{\mu_k}[\|\hat{g}_k\|_2^2] \tag{84}$$
$$\le \eta_1\tau\mathbb{E}_{\mu_k}[\langle\theta_k, g_k\rangle] + \eta_1\tau^2 d + \frac{\tau}{2}\eta_1^2 R_{g4}^{1/2}.$$

And for second order part,

$$\mathbb{E}\left[\mathbb{E}_{\mu_k \otimes \rho}\left[\left\langle \nabla^2\frac{\delta J}{\delta\mu}[\mu_k, \mathcal{K}_*^+[\mu_k]], (\eta_1 g_k + \sqrt{2\eta_1\tau}\xi_k^1)^{\otimes 2}\right\rangle\right]\bigg|\mathscr{H}_k\right]$$
$$\le M_2\eta_1^2\mathbb{E}_{\mu_k}[\|\hat{g}_k\|_2^2] + 2\eta_1\tau\mathbb{E}_{\mu_k}\left[\mathrm{Tr}\left(\nabla^2\frac{\delta J}{\delta\mu}[\mu_k, \nu_k]\right)\right] \tag{85}$$
$$= M_2\eta_1^2\mathbb{E}_{\mu_k}[\|\hat{g}_k\|_2^2] - 2\eta_1\tau\mathbb{E}_{\mu_k}[\langle\hat{g}_k, \nabla\log\mu_k\rangle] + 2\eta_1\tau^2 d$$
$$\le M_2\eta_1^2 R_{g4}^{1/2} - 2\eta_1\tau\mathbb{E}_{\mu_k}[\langle g_k, \nabla\log\mu_k\rangle] + 2\eta_1\tau^2 d$$

Hence, we can get new constants as in (17). Substitute into Theorem 1 we can get a result with remainder $R_2$. $\square$

## D. Proofs for Section 4

*Proof of Proposition 3.* Let $\nu' \propto \exp(-\frac{\|\omega\|^2}{2})$. By the Bakry-Émery condition (Bakry & Émery, 2006), $\nu'$ satisfies LSI with parameter 1. Then according to the Holley-Stroock perturbation principle (Holley & Stroock, 1986), if $\nu'$ and $\nu$ are Gibbs measures with Hamiltonian $H$ and $\phi + H$, respectively, and $\nu'$ satisfies LSI with parameter $\alpha'$, then $\nu$ also satisfies LSI with parameter $\alpha \ge e^{-(\sup\phi - \inf\phi)}\alpha'$. Note that the first variation of $J$ as defined in (10) equals

$$\frac{\delta J}{\delta\nu}[\mu, \nu](y) = \mathbb{E}_\mu[G(\theta, \omega)]$$

Then the Gibbs functional becomes

$$\mathcal{K}_\mu^+[\nu] = \exp\left(\tau^{-1}\mathbb{E}_\mu[G(\theta,\omega)] - \frac{\|\omega\|^2}{2}\right).$$

Let

$$H = \frac{\|\omega\|^2}{2}, \ \phi(\omega) = \tau^{-1}\mathbb{E}_\mu[G(\theta,\omega)].$$

We can verify that $\sup\phi - \inf\phi = 2\tau^{-1}G_0$. This implies that $\mathcal{K}_\mu^+[\nu]$ satisfies LSI with parameter $\alpha = \frac{1}{\exp(2G_0\tau^{-1})}$. We can show the same conclusion for $\mathcal{K}_\nu^-[\mu]$ using a similar technique, and we omit here for brevity.

Then we can verify the conditions in Assumption 2. Observe that the Wasserstein gradient of (10)

$$\nabla_\theta \frac{\delta J}{\delta\mu} = \int \nabla_\theta G(\theta,\omega)\mathrm{d}\nu, \quad \nabla_\omega \frac{\delta J}{\delta\nu} = \int \nabla_\omega G(\theta,\omega)\mathrm{d}\mu \tag{86}$$

are both smooth up to fourth order since $\|\nabla^i G(x,y)\|_F \le G_i, i = 0, 1, \ldots, 4$. Moreover, since $J$ is bilinear,

$$J(\mu',\nu) - J(\mu,\nu) = \mathbb{E}_{\delta\mu}\left[\frac{\delta J}{\delta\mu}[\mu,\nu]\right], J(\mu,\nu') - J(\mu,\nu) = \mathbb{E}_{\delta\nu}\left[\frac{\delta J}{\delta\nu}[\mu,\nu]\right]. \tag{87}$$

Hence, the second-order term equals zero. Meanwhile, the cross second-vairation $\frac{\delta^2 J}{\delta\mu\delta\nu} = G(\theta,\omega)$ satisfies boundedness, which certificating Assumption 2. $\qquad\square$

*Proof of Proposition 4.* Similar to proof of Proposition 3, note that the first variation of $J$ as defined in (12) equals

$$\frac{\delta J}{\delta\nu}[\mu,\nu] = \mathbb{E}_\mathcal{D}[(F_y'(\mathbb{E}_\mu[\sigma_1(x;\theta)], \mathbb{E}_\nu[\sigma_2(y;\omega)])\sigma_2(y;\omega)].$$

Then the Gibbs functional becomes

$$\mathcal{K}_\mu^+[\nu] = \exp\left(\tau^{-1}\left(\mathbb{E}_\mathcal{D}[(F_y'(\mathbb{E}_\mu[\sigma_1(x;\theta)], \mathbb{E}_\nu[\sigma_2(y;\omega)])\sigma_2(y;\omega)]\right) - \frac{\|\omega\|^2}{2}\right).$$

Let

$$H = \frac{\|\omega\|^2}{2}, \ \phi(\omega) = \mathbb{E}_\mathcal{D}[\tau^{-1}(F_y'(\mathbb{E}_\mu[\sigma_1(x;\theta)], \mathbb{E}_\nu[\sigma_2(y;\omega)])\sigma_2(y;\omega))].$$

We can verify that $\sup\phi - \inf\phi = 2\tau^{-1}F_1 m_0$. This implies that $\mathcal{K}_\mu^+[\nu]$ satisfies LSI with parameter $\alpha = \frac{1}{\exp(2F_1 m_0\tau^{-1})}$. We can show the same conclusion for $\mathcal{K}_\nu^-[\mu]$ using a similar technique, and we omit here for brevity.

Then, we will verify $J$ defined in (12) satisfies Assumption 2.

It is evident that $J$ is convex in $\mu$ and concave in $\nu$. Furthermore, we note that

$$\left\|\nabla^i \frac{\delta}{\delta\mu} J(\mu,\nu)\right\|_F = \mathbb{E}_\mathcal{D}[F_x'(\mathbb{E}_\mu[\sigma_1(x;\theta)], \mathbb{E}_\nu[\sigma_2(y;\omega)])\|\nabla^i\sigma_1(x;\theta)\|_F] \le F_1 m_i. \tag{88}$$

And the same for $\|\nabla^i \frac{\delta}{\delta\nu} J(\mu,\nu)\|_F$.

For second variation term, note that

$$\left\|\frac{\delta^2}{\delta\mu\delta\nu} J(\mu,\nu)\right\|_\infty = \mathbb{E}_\mathcal{D}[F_{xy}''(\mathbb{E}_\mu[\sigma_1(x;\theta)], \mathbb{E}_\nu[\sigma_2(y;\omega)])\|\sigma_1(x;\theta)\|_\infty\|\sigma_2(y;\theta)\|_\infty] \le Lm_0^2. \tag{89}$$

Similarly,

$$\left\|\nabla_\theta\nabla_\omega^\top \frac{\delta^2}{\delta\mu^2} J(\mu,\nu)\right\|_F = \mathbb{E}_\mathcal{D}[F_{xy}''(\mathbb{E}_\mu[\sigma_1(x;\theta)], \mathbb{E}_\nu[\sigma_2(y;\omega)])\|\nabla\sigma_1(x;\theta)\|_F\|\nabla\sigma_2(y;\theta)\|_F] \le Lm_1^2 \tag{90}$$

And similar for $\left\|\nabla_\theta\nabla_{\theta'}^\top \frac{\delta^2}{\delta\mu^2} J(\mu,\nu)\right\|_F$, $\left\|\nabla_\omega\nabla_{\omega'}^\top \frac{\delta^2}{\delta\nu^2} J(\mu,\nu)\right\|_F$. Hence we has verified all the condition in Assumption 2. $\qquad\square$

