# OpenReview forum: "Convergence of Mean-Field Langevin Stochastic Descent-Ascent for Distributional Minimax Optimization"
_ICML.cc/2025/Conference — ICML 2025 spotlightposter_

### Official Review · Reviewer_5VGF · 2025-03-13

**Overall Recommendation:** 3

**Summary:**

This paper studies the mean-field Langevin (stochastic) descent-ascent (MFL-DA) algorithm for solving distributional minimax optimization problems. The authors demonstrate that the infinite-particle limit of discrete-time MFL-DA is able to converge to the unique stationary point of the problem with a convergence rate of $\frac{1}{\epsilon}\log\frac{1}{\epsilon}$ measured in squared 2-Wasserstein distance. The authors show applications of their result to finding mixed Nash equilibria of zero-sum games, generative adversarial networks, and mean-field neural networks.

**Claims And Evidence:**

The claims seem clear and well-supported.

**Essential References Not Discussed:**

Most essential references are discussed in the paper. I think the authors can also discuss [1] where the mean-field Langevin algorithm is used for optimization over signed measures. Specifically, that paper contains ideas for going beyond pessimistic LSI constant estimates which can be useful for the results here as well, and similar to this paper, they also need a two-timescales approach for their analysis.

[1] G. Wang, A. Mousavi-Hosseini, L. Chizat. "Mean-Field Langevin Dynamics for Signed Measures via a Bilevel Approach." NeurIPS 2024.

**Experimental Designs Or Analyses:**

Not applicable.

**Methods And Evaluation Criteria:**

Not applicable since this is a theory paper.

**Other Comments Or Suggestions:**

Please see below.

**Other Strengths And Weaknesses:**

**Strengths**:

Solving minmax optimization on the space of distributions is a fundamental problem, and it is nice to have discrete-time convergence guarantees for the mean-field Langevin descent-ascent algorithm. Also, the paper is mostly well-written and easy to ready.

**Weakness**:

* A main concern for me is that there is almost no discussion on the role of other parameters besides $\epsilon$ in the convergence rate. A major weakness of this type of mean-field Langevin analysis is that convergence requires $\tau$ to be at least linearly small with ambient dimension $d$. When plugged into the pessimistic LSI bound, this implies $\alpha$ that is exponentially small in $d$, and thus a convergence rate that is exponentially large in $d$. This is a drawback of this type of analysis and not a weakness of this paper in particular, but I think it is better to be explicitly discussed.

* Similarly, the role of other parameters is not clear/made explicit. For example, is it better to have $\eta_1 \gg \eta_2$ or $\eta_1 \ll \eta_2$? How do quantities like $\alpha$, $\sigma^2$, and $\zeta$ enter the final convergence rate? While these questions might be answered by following certain quantities in the appendix, I think it would be nice to have summaries of the convergence rate in the main text.

* The convergence analysis yields a bound on $\mathcal{L}$ and the Wasserstein distance to $\mu^*$ and $\nu^*$. Can such bounds be turned into a bound on the suboptimality $E(\mu_K,\nu_K) - E(\mu^*,\nu^*)$? I am asking this in particular since bounding this suboptimality is possible for distributional minimization problmes with mean-field Langevin.

* Additional suggestions are discussed below.

**Questions For Authors:**

1. It seems that $\frac{1}{\alpha_1}$ should be replaced with $\alpha_1$ in Corollary 1, otherwise we can drive the Wasserstein distances to zero by simply letting $\alpha_1 \to 0$.

2. What is the optimal value of $\lambda$ for Theorems 1, 2, and Corollary 1?

3. Does Assumption 4 hold point-wise for all $\theta$ and $\omega$? I believe the expectation is over the random noise for estimating $g_k$ and $h_k$, which still leaves out $\theta$ and $\omega$.

4. Do we expect the constants that appear in the paper, such as those in Assumptions 1-4 and in Proposition 3 to be dimension-dependent in typical settings?

5. Equation (8) seems to have a typo, why are you using the expectation notation?

6. Typo in the first line of Equation (9), I believe $\nu$ should be $\nu’$.

**Relation To Broader Scientific Literature:**

This paper contributes to a long line of work of applications of the mean-field Langevin dynamics for optimization in the space of probability distributions. As discussed in the paper, this is (to my knowledge) the first discrete-time convergence guarantee for mean-field Langevin descent-ascent for solving distributional min-max problems. Both the algorithm and the problem are of significant interest to the community.

**Theoretical Claims:**

I did not go over the details of the proof but the overall claims seem to be consistent with the literature and the proof technique seems sound from a high level.

---

> ### Author Rebuttal · Authors · 2025-03-29
>
> Thank you for your thoughtful and detailed feedback! Below, we address the comments and questions point-by-point.
>
> **Pessimistic LSI bound**:  We will explicitly comment on the weakness of this type of analysis, and we are trying to overcome this in our ongoing research.
>
> **Explicit dependence on other parameters**:
> In the revised version, we will include a dedicated paragraph discussing the roles of parameters and here we provide a sketch.
>
> **On dimension $d$ and regularization parameters**: We agree that a more explicit discussion of the role of parameters in the convergence rate would be beneficial. Viewing $\tau,\eta_1,\eta_2$ as small numbers, in Appendix A.3 on page 11, the remainder $r_{g2}, r_{h2}$ due to the second moment are $\max\\{O(1),\tau d\\}$, similarly $r_{g3}, r_{h3}$ are $\max \\{ O(1),\tau^{3/2}d^{3/2} \\} $. Substituting them into $\Gamma_0, \Gamma_1, \Gamma_2$, we can get $\Gamma_0=\max\\{O(1), \tau d,\tau^{2}d^{2},\frac{d}{\alpha^{1/2}\tau} \\}$, and $\Gamma_{1(2)}=\max\\{O(1),\tau d,\tau^2d^2\\}$.  Substutiuting them into $R_1$ in (10), we can get $R_1=O(\frac{d^2\eta_1}{\tau^3\alpha^3})$. Let $R_1=\epsilon$ and choose $\eta_1= O(\frac{\epsilon \tau^3 \alpha^3}{d^2})$, we obtain a sample complexity $K=O(\frac{d^2}{\epsilon \tau^4 \alpha^4}\log \frac{1}{\epsilon})$. Comparing with sample complexity $K=O(\frac{d^2}{\epsilon \tau^2 \alpha^2}\log \frac{1}{\epsilon}$) in [1] for the MFLD of minimization problems, the higher orders of $\tau, \alpha$ is because the two-timescale algorithm MFL-SDA: the overall complexity of the algorithm depends on the slower descent step.
>
> **On other parameters:** The variance $\zeta,\sigma^2$ only affect the $O(1)$ term in $\Gamma_i$. The parameters $\eta_1,\eta_2$ correspond to the fast descent and fast ascent regime similar to [3], and thus their choice depends on the instance and the user's emphasis on the descent/ascent part.
>
> **Another standard of suboptimality**: Our problem is to find a saddle point rather than a maxima(mimima), hence  $E(\mu_K,\nu_K)-E(\mu^*,\nu^*)$ may not apply to our analysis.
>
> ## Addressing References Not Discussed:
>
> Thank you for pointing this out! We will cite this paper in the revision and investigate if their Theorem 5.2 can inspire a better LSI bound for our problem.
>
> ## Addressing Questions:
>
> **A1.** Thank you for catching this: the Talagrand's inequality should be $W_2^2(\mu_k,\nu^*)\leq \frac{2}{\alpha}\mathrm{KL}(\mu_k|\mu^*)$.
>
> **A2.** $\lambda$ corresponds to the Lyapunov function similar to [2] [3], and it is not an explicit hyperparameter in our algorithm.
>
> **A3.**  Yes, it holds point-wise for all $\theta$ and $\omega$.
>
> **A4.** See our response on the dimension above.
>
> **A5, A6.** Thank you for noticing this. We will correct the notation.
>
> We thank you very much again for the constructive feedback and helpful suggestions!
>
> ### Reference:
>
> [1] Nitanda, A., Wu, D., & Suzuki, T. (2022). Convex analysis of the mean field Langevin dynamics. In International Conference on Artificial Intelligence and Statistics (pp. 9741-9757). PMLR.
>
> [2] Lu, Y. (2023). Two-scale gradient descent ascent dynamics finds mixed Nash equilibria of continuous games: A mean-field perspective. In International Conference on Machine Learning (pp. 22790-22811). PMLR.
>
> [3] Yang, J., Kiyavash, N., & He, N. (2020). Global convergence and variance-reduced optimization for a class of nonconvex-nonconcave minimax problems. arXiv preprint arXiv:2002.09621.

---

### Official Review · Reviewer_GF3y · 2025-03-13

**Overall Recommendation:** 3

**Summary:**

This paper studies the convergence rate of discrete-time mean-field Langevin stochastic descent-ascent for min-max problems in distributional optimization under log-Sobolev inequality condition. The authors claim that the derived convergence rate is near-optimal compared to its Euclidean counterpart. The paper includes two examples: zero-sum games and mean-field neural networks. However, no experimental results are provided.

**Claims And Evidence:**

The theoretical results are consistent with the authors' claims.

**Essential References Not Discussed:**

No

**Experimental Designs Or Analyses:**

Not applicable.

**Methods And Evaluation Criteria:**

Not applicable.

**Other Comments Or Suggestions:**

In Eq (9):

L1(u) = max E(u,v') - minmax E(u',v')
L2(u,v)=max E(u,v') - E(u,v)

**Other Strengths And Weaknesses:**

Strengths:
1. The topic is interesting.
2. The paper is easy to read.

Weaknesses:
1. The analysis and results appear elementary and straightforward, closely following previous works such as Kim et al. (2024), Yang et al. (2020), Chen et al. (2022), Suzuki et al. (2023), and Nitanda et al. (2022). Specifically, the variable lambda in the Lyapunov function seems unnecessary, as it can be set to 1 in this case. Given this, the results appear quite direct based on current results in MFLD, limiting the paper’s technical contribution.

2. What are the advantages of the MFL-SDA algorithm compared to MFL-AG and MFL-ABR? The lack of comparison makes the motivation unconvincing. A discussion of their relative strengths and weaknesses would improve clarity.

3. The two provided examples are restricted to two-layer neural networks, and no experimental evidence is given. I would expect more interesting applications and empirical results to strengthen the paper’s impact.

**Questions For Authors:**

See my comments in weaknesses.

**Relation To Broader Scientific Literature:**

The topic is important to optimization theory and has significant relevance to the machine learning community

**Theoretical Claims:**

I did not verify the details of the proofs.

---

> ### Author Rebuttal · Authors · 2025-03-30
>
> Thank you for your thoughtful comments! Below, we respond to the concerns and clarify the novelty and contributions of our work.
>
>
> **On our analysis**:   We would clarify that both our proof technique and the resulting conclusions differ from the aforementioned papers in several fundamental ways. First, these prior works (*Kim et al. (2024)*, *Chen et al. (2022)*, *Suzuki et al. (2023)*, *Nitanda et al. (2022)*) tackle the discrete-time MFLD problem by first establishing convergence in continuous time (typically via Wasserstein gradient flow) and then controlling the discretization error as a separate step. In contrast, our proof directly analyzes the per-step improvement using Taylor expansion and remainder estimates. This not only provides a more elegant and adaptable proof framework, but also yields stronger results as detailed in the next point.
>
> Moreover, our results also differ in several key aspects. *Kim et al. (2024)* do not study MFL-SDA. To our best knowledge, their analysis of other algorithms does not extend to our established convergence rate of MFL-SDA. While *Yang et al. (2020)* study a minimax optimization in Euclidean space, and our analysis shares some structural similarities with theirs, the time-discretization error anlaysis of the probability functional on distributional space requires substantially more sophistication -- we devote pages of proofs to controlling this error, whereas in the Eucliean case, it follows directly from smoothness assumptions. The other cited references--*Chen et al. (2022)*, *Suzuki et al. (2023)*, and *Nitanda et al. (2022)*--focus on minimization problems, whereas minimax problems present more complexity due to the interaction between two players.
>
> **On the Lyapunov function**: As you said, $\lambda$ can be fixed to 1. Yet, our chosen form of the Lyapunov function follows what is commonly used in previous works on minimax problems in Euclidean space (*Yang et al. (2020)*) and in continuous-time MFLD (*Lu (2023)*).
>
> **Comparison with MFL-AG and MFL-ABR:** First, our analysis of stochastic gradient descent-ascent with last-iterate convergence and the inexact gradient analysis closely aligns with practical implementations.
> Second, MFL-AG achieves a sample complexity $O(\epsilon^{-O(1/\alpha)})$ to reach an $O(\epsilon)$-approximation of the saddle point, where the bound on LSI constant $\alpha$ can be small--potentially even of order $1/(2+d/2)$. In contrast, our convergence analysis for MFL-DA establishes a sample complexity $O(\frac{1}{\epsilon}\log\frac{1}{\epsilon})$. MFL-ABR is a double-loop algorithm, it is shown that the outer loop has an $O(\frac{1}{\epsilon}\log\frac{1}{\epsilon})$ sample complexity; however, it needs an inner loop and the total complexity is not specified.
>
> **On Experimental Results and Applications:**
> The primary aim of our work is to establish convergence guarantees for commonly used stochastic gradient descent-ascent algorithms. We acknowledge the value of empirical validation, we have included a numerical experiment based on Example 2 in our paper. We apply our algorithm to the nonlinear instrumental variable (NPIV) regression problem $$
> \min_f \max_g E[g(Z)(Y-f(X))-\frac{1}{2}g(Z)^2+\lambda R(f)]
> $$
> where $R$ is a regularizer.
> Using the problem setup and datasets in [1,2], we compare our algorithm with a classic series-approximation (SA) method [3] based on out-of-sample MSE (lower values indicate better performance) and average R² (higher values indicate better performance)
>
> *Engel Curve*
> | Method                    | Average MSE | Average R² |
> | ------------------------- | ----------- | ---------- |
> | MFL-SDA                 | **0.00698**  | **0.256**   |
> |SA | 0.00730  | 0.218   |
>
> *Returns to schooling*
> | Method                    | Average MSE | Average R² |
> | ------------------------- | ----------- | ---------- |
> | MFL-SDA                 | **0.1494**  | **0.0799**   |
> | SA | 0.1626  | 0.1543   |
>
> **4. Clarification of Equation (9):**
> Good catch! We will revise Equation (9) in the updated version.
>
> Thank you again for your feedback, and we hope our response clarifies the contribution of our paper.
>
> ### References:
> [1]Hausman, J. A., Newey, W. K., & Powell, J. L. (1995). Nonlinear errors in variables estimation of some Engel curves. *Journal of Econometrics*, *65*(1), 205-233.
>
> [2] Card, David. Estimating the return to schooling: Progress on some persistent econometric problems. *Econometrica* 69.5 (2001): 1127-1160.
>
> [3] Newey, Whitney K., and James L. Powell. Instrumental variable estimation of nonparametric models. *Econometrica* 71.5 (2003): 1565-1578.

---

### Official Review · Reviewer_YCRA · 2025-03-13

**Overall Recommendation:** 4

**Summary:**

This paper analyzes a natural algorithm for distribution min-max optimization, which consists in taking alternating Langevin steps. The main contribution of the paper is theoretical analysis of this algorithm for the case where the gradients are exact as well as the case where the gradients are in-exact. Their analysis avoids the more standard approach which first analyzes the continuous time flow and then bounds the discretization error. The main application is to mean-field networks.

**Claims And Evidence:**

I am seriously concerned about their claim, after Theorem 1, that the bias term $R_1$ is $O(\eta_1)$. In fact, when I examine the definitions of $\Gamma_1, \Gamma_2, \Gamma_3$, it looks to me that they are in fact $O(1/\eta_1)$ (this is coming from the final terms in the definitions of $r_{g2}, r_{h2}$). Unless there is a typo, this would seem to indicate that the bias term $R_1$ is in fact $O(1)$, meaning that their main result only shows that the objective doesn't increase more than $O(1)$ throughout the trajectory. I assume that this problem extends to Theorem 2, but the relevant constants don't seem to be defined in Section A.3 (where are they?).

Another -- although less critical -- issue is Assumption 3 on the log-Sobolev constants. Although they consider several applications of their results, they don't demonstrate any settings where their assumptions hold. Of course, verifying their assumptions for neural networks is likely far too difficult, but as a sanity check, I would like to seen an example of a setting where Assumption 3 actually holds. I think this would make the paper stronger.

Finally, I am probably confused, but in Corollary 1, I don't understand why there must be unique $\mu^*, \nu^*$ that the algorithm even converges to? What am I missing here?

**Essential References Not Discussed:**

None that I am aware of.

**Experimental Designs Or Analyses:**

There were no experiments but I don't think that's necessary.

**Methods And Evaluation Criteria:**

Yes.

**Other Comments Or Suggestions:**

- I don't follow equation 24.
- Equation 46 uses $\tilde \mu$ without defining it (reading onwards I suppose that it is meant to be a distribution coming from the mean-value theorem).

**Other Strengths And Weaknesses:**

If the bias term were truly $O(\eta_1)$ the paper would be a strong theoretical contribution. In fact, it would probably even imply a new analysis of discrete-time Langevin dynamics since that seems to be a special case of their setup. But this is exactly why I am a bit skeptical of their results -- it's hard to imagine that they found a completely new analysis of Langevin dynamics that simulatenously holds in their more general setup. If the bias term is indeed only $O(1)$ then I don't think the paper is strong enough, as it merely asserts the objective doesn't blow up.

**Questions For Authors:**

Please address the issue about the size of the bias term $R_1$.

**Relation To Broader Scientific Literature:**

The paper is trying to push forward on discrete time analysis of distributional min-max problems. This is certainly a worthy problem, and it seems their contribution would be novel. However, I am seriously concerned about the bias terms.

**Theoretical Claims:**

I checked some of the proofs in the appendix, but not all of them and not in full detail. I didn't find any serious issues, and at a high level the results seem plausible. The main concern is the dependence of the bias term on the step-sizes $\eta_1, \eta_2$, see above.

---

> ### Author Rebuttal · Authors · 2025-03-30
>
> Thank you very much for your critical feedback, especially your sharp observations about the order of the bias term. Below, we address your concerns.
> $\newcommand{\bE}{\mathbb{E}} \newcommand{\o}{\omega}$
> ## Addressing Weaknesses
>
> **Bias term of second moment**: Indeed, the norm control in the submitted version (adopted from [3, Lemma 1]) is sub-optimal. Below, please find a stronger control, adopted from [2, Lemma 1]. The corrected expression for $r_{g2}$ should be $r_{g2}= 2M^2_1+2\tau^2\bE_{\mu_0}[\\|\\theta_0\\|^2_2]+4M^2_1+ \tau(4\eta_2M^2_1+4\tau d)$ and similarly for $r_{h2}$. They are of order $O(1)$, thereby the bias term $R_1$ in Theorem 1 is $O(\eta_1)$.
>
> $$
> \begin{aligned}
>   &\bE_{\nu_{k+1}}[\\|\o_{k+1}\\|^2]\\\\
>   =&\bE_{\nu_k\otimes \rho}[\\|\o_k+\eta_2 h_k+\sqrt{2\eta_2\tau}\xi^2_k\\|^2]\\\\
>   =&\bE_{\nu_k}[\\|\o_k\\|^2]+2\bE_{\nu_k\otimes \rho}[\langle \o_k,\eta_2 (h_k+\tau\o_k-\tau\o_k)+\sqrt{2\eta_2 \tau}\xi^2_k\rangle]+\bE_{\nu_k\otimes \rho}[\\|\eta_2h_k+\sqrt{2\eta_2 \tau} \xi^2_k\\|^2]\\\\
>    \leq&\bE_{\nu_k}[\\|\o_k\\|^2] + 2\eta_2 M_1 \bE_{\nu_k}[\\|\o_k\\|]-2\eta_2\tau \bE_{\nu_k}[\\|\o_k\\|^2]+2\eta^2_2( M^2_1 + \tau^2 \bE_{\nu_k}[\\|\o_k\\|^2]) +2\eta_2\tau d\\\\
>   \leq& (1-2\eta_2\tau +\frac{\eta_2\tau}{2}+2\eta^2_2\tau^2) \bE_{\nu_k}[\\|\o_k\\|^2] + 2\eta_2 M^2_1/\tau +2\eta^2_2 M^2_1 + 2\eta_2\tau d\\\\
>   \leq& (1-\eta_2\tau)\bE_{\nu_k}[\\|\o_k\\|^2] +\eta_2(2M^2_1/\tau+2\eta_2M^2_1+2\tau d),
> \end{aligned}
> $$
> where the first equality uses the fact that $\xi_k^2$ is an independent zero-mean normal; the first inequality follows the fact that
> $\\|h_k+\tau\o_k\\| = \\|\nabla \frac{\delta J}{\delta \nu} \[\mu_{k+1}, \nu_k\](\o_k)\\| \leq M_1$; the second inequality is because $2M_1\bE_{\nu_k}[\\|\\o_k\\|]\leq \frac{\tau}{2} (\bE_{\nu_k}[\\|\o_k\\|])^2 + 2M^2_1/\tau \leq  \frac{\tau}{2}\bE_{\nu_k}[\\|\o_k\\|^2]+2M^2_1/\tau$ and last inequality holds for $\tau<1/(4\eta_2)$. Hence, we obtain
> $$
> \begin{aligned}
> \bE_{\nu_k}[\\|\o_k\\|^2] & \leq  (1-\eta_2\tau)^k \bE_{\nu_0}[\\|\o_0\\|^2]+ 2\eta_2 (M_1^2/\tau + \eta_2 M_1^2 + \tau d )\sum_{j=0}^{k-1}(1 - \eta_2\tau)^j\\\\
> &\leq \bE_{\nu_0}[\\|\o_0\\|^2]+  \frac{2(M^2_1/\tau + \eta_2 M^2_1 + \tau d)}{\tau},
> \end{aligned}
> $$
> The remaining part of the proof stays the same.  Hence, we can directly check that the remaining constants $r_{gi}$, $r_{hi}$ as well as $\Gamma_0$, $\Gamma_1$, $\Gamma_2$ are all of order $O(1)$. Since $\eta_1$ and $\eta_2$ are of the same order, we conclude that the bias term $R_1= \frac{\lambda(\Gamma_2\eta_2^2+(1-2\eta_2\tau\alpha)(\Gamma_1+\Gamma_0)\eta_1^2)+\Gamma_1\eta_1^2}{\eta_1\tau\alpha}$ is of order $O(\eta_1)$.
>
> Regarding your question on the definition of constants in (18), they are defined in the proof of Theorem 2 (Line 1214, etc.). We apologize for not explicitly displaying them.
>
> **Assumptions of LSI in examples**: Indeed, we have provided two applications in Section 4, with the LSI being verified in Proposition 3 and Proposition 4, respectively. This follows a line of work like [1, proposition 5.1] and [2, Appendix A]. The setups therein are satisfied in our context.
>
> **Existence and uniqueness of optimal $\mu^\*,\nu^\*$ :**  The primal objective $J(\mu, \nu)$ is convex in $\mu$ and concave in $\nu$. With an additional KL (or entropy) regularization, the regularized objective $E(\mu, \nu)$ becomes strongly convex in $\mu$ and strongly concave in $\nu$. This ensures the existence and uniqueness of the mixed Nash equilibrium $(\mu^*, \nu^*)$; please refer to the detailed discussion before equation (3) in the manuscript as well as [2, Proposition 2.1].
>
> ## Addressing Questions
> **A1**. To get (24), the first line follows directly from equation (23); the second line follows from the definition of $T$; the third line follows from the Taylor's expansion of $\log {\rm det}$, where $\bar{\o}_k$ is the point that achieves the mean-value; and the fourth line follows from the property of trace operator and that ${\rm Tr}(h_k^2) =\\|\nabla h_k\\|_F^2 = (\\|\nabla^2 \frac{\delta J}{\delta \nu}\\|_F - \tau)^2 \leq (M_2 + \tau)^2$.
>
> **A2**. Yes, you're absolutely right.
>
> Thank you so much again for your insightful and constructive feedback. We hope our responses alleviate your concerns.
>
> ### References:
> [1]Chizat, L. Mean-field Langevin dynamics: Exponential convergence and annealing. *Transactions on Machine Learning Research*, 2022.
>
> [2]Suzuki, T., Wu, D., and Nitanda, A. Mean-field Langevin dynamics: Time-space discretization, stochastic gradient, and variance reduction.  *Advances in Neural Information Processing Systems*, 36, 2024.
>
> [3] Nitanda, Atsushi, Denny Wu, and Taiji Suzuki. Convex analysis of the mean field langevin dynamics. *International Conference on Artificial Intelligence and Statistics*, 2022.

---

> > ### Comment · Reviewer_YCRA · 2025-04-07
> >
> > Thank you for your response, it does seem like this addresses the order of the bias issue. I also apologize for missing Propositions 3 and 4. I have updated my score accordingly.

---

> > > ### Author Response · Authors · 2025-04-08
> > >
> > > Thank you very much for your thoughtful comments and for taking the time to revisit your evaluation. We're glad to hear that the updated version addresses the issue regarding the order of the bias.

---

### Official Review · Reviewer_eWwv · 2025-03-16

**Overall Recommendation:** 3

**Summary:**

The paper analyzes a Langevin-type scheme for finding equilibria in mean-field games under convexity and smoothness assumptions. The rates obtained scale as $\widetilde{O}(1/\varepsilon)$, which agrees with the rate in Euclidean space. An extension to stochastic gradients is also considered.

**Claims And Evidence:**

The main selling point of this work is the $\widetilde{O}(1/\varepsilon)$ rate, under standard assumptions on the boundedness of (derivatives of) differentials of the objective. This result is state-of-the-art compared to prior work (which generally has higher complexity). There is also an extension to stochastic gradient oracles when the oracle has bounded error in the second and third moments. The authors provide solid proofs for their claims.

**Essential References Not Discussed:**

As far as I am aware, the most relevant references have been covered.

**Experimental Designs Or Analyses:**

The paper is theoretical in nature and therefore does not contain an experimental component. However, some ramifications for typical applications are discussed.

**Methods And Evaluation Criteria:**

This is not applicable to this paper.

**Other Comments Or Suggestions:**

Line 143: “We set the scaling factor the weight decay the regularization” does not make sense, please revise.

Equation 9 exceeds the column spacing. This occurs in multiple other places; please fix this.

**Other Strengths And Weaknesses:**

**Strengths:**

I believe this work is impactful as it makes a significant improvement to the rate estimate for an important statistical problem.

The rates are examined in the context of various applications, with interpretable and clear results.

**Weaknesses:**

The result is mainly theoretical, as in general a particle-algorithm is needed for implementability (which complicates the rate). This is agreed upon by the authors; it is unclear to me whether the analysis scheme in this paper can be preserved after particle discretization.

It would be helpful if the authors included other important parameters in their rate estimates, such as the dimensions.

**Questions For Authors:**

What is the dependence of the result on the dimension, and the condition numbers (Hessian bounds, etc.)? How does it compare to prior work?

It is a bit strange that third-moment boundedness is needed for the stochastic gradient oracle, when normally it is only the second moment that is required. Can the authors comment if they believe this requirement is fundamental?

Although the paper attains a sharp rate, this is for the mean-field version of the algorithm. It is not clear whether this type analysis can be carried out using this framework for a finite particle algorithm (accounting for additional error of the particles). Can the authors comment?

**Relation To Broader Scientific Literature:**

The paper is related to prior work on mean-field games, in particular to Langevin-type algorithms for computing equilibrium points. The main claim of this paper is that it improves the rates compared to those prior works (at least with respect to the inverse accuracy dependence). Earlier work had a super-linear dependence on $\varepsilon$ or required an inner-outer loop complexity.

**Theoretical Claims:**

The results make sense and the proofs appear rigorous; I checked all the results (without verifying all technical details).

---

> ### Author Rebuttal · Authors · 2025-03-29
>
> Thank you very much for your thorough and constructive feedback. Below, we address your main points raised.
> $\newcommand{\bs}{\boldsymbol}$
> ## Addressing Weakness:
>
> **Particle Discretization**:  For this problem, we have verified that our method is feasible in the particle setting. As an illustration, we briefly summarize the adjustments needed in our proof technique under the particle approximation setting:
>
> We set the joint distribution $\bs\mu,\bs\nu$ as $N$ finite particles $\bs\theta=[\theta^i]^N_{i=1}$ and $\bs \omega = [\omega^i]^N_{i=1}$. In this case, the LSI has the form adapted from [1, Lemma 7]
>
> $$
>   \frac{\tau^2}{N} \sum^N_{i=1} \mathbb E_{\bs\nu_k}[\\|h_i(\bs\omega_k)-\nabla \log \bs\rho_k(\bs\omega_k)\\|^2_2]\geq 2\alpha \tau {\mathcal L_2^N}(\bs\mu_{k+1},\bs\nu_k)+O(\frac{1}{N})
> $$
> where $\bs\mu_k$ and $\bs\nu_k$ denote the distribution of $\bs\theta$ and $\bs\omega$ in the $k$-th iteration, respectively; $h_i$ denotes the partial derivative w.r.t the $i$-th particle; $\mathcal{L}_2^N$ denotes the Lynapunov function for the counterpart on the product space; $\bs\rho$ denotes the Gibbs distribution; $O(\frac{1}{N})$ is from the propagation of chaos. Other log-Sobolev inequalities used in our setting have similar forms. Thereby, most derivations in the manuscript involving Taylor expansions can be extended to analyze the joint distribution of particles, but with additional approximation error $O(1/N)$ due to weak particle interactions. The per-step improvement would be of the form
>
> $$
> \mathcal L(\bs\mu_{k+1},\bs\nu_{k+1})\leq (1-2\eta_1\tau\alpha)\mathcal L(\bs\mu_{k},\bs\nu_{k}) + O(\eta_1^2) + O(\frac{\eta_1}{N})
> $$
>
> where the $O(\frac{\eta_1}{N})$ term comes from the propagation of chaos. Building on our current line of thought, we believe this approach can be extended to the stochastic gradient setting while maintaining the same sample complexity order as well. However, due to the extensive computations involved, we will present a complete proof of our results in an upcoming extended journal version.
>
> **Parameters**: Due to the space limit, please refer to our response to Reviewer 5VGF for a detailed analysis of how the sample complexity in Theorems 1 and 2 depends on various parameters, including the dimension $d$. Notably, the sample complexity of MFL-AG in [2] is $O(\epsilon^{-O(1/\alpha)})$, which is exponential in the LSI constant $\alpha$, hence in $d$ under their conservative bound $O(1/(2+d/2))$ of $\alpha$, whereas our bound does not suffer from this exponential dependence. The Hessian bound $C_1$ occurs in bias $\Gamma_i=O(C_1)$. As for the condition number, adopting the notion of effective condition number in [3, Theorem 2.1], our bound has a similar dependence in the case of zero-sum game.
>
> ## Addressing Questions:
>
> **Line 143 and Equation (9)**: Thank you for pointing this out. We will correct the issues in the updated version of the manuscript.
>
> **Boundedness of third-moment**: The existence of this issue arises from the presence of $\mathbb{E}_{\mu_k}[\\|g_k\\|_2^3]$ in our remainder term. Therefore, in the stochastic gradient part, to ensure that this remainder term remains uniformly bounded, we need to impose a bound on the third-order moment. In fact, [1] assumed that the stochastic gradient is uniformly bounded with respect to any $\omega$ and $\theta$, which is stronger than ours.
>
> Thank you again for your valuable feedback! We will correct the typos and formatting issues as you suggested.
>
> ### References
>
> [1] Suzuki, T., Wu, D., & Nitanda, A. (2023). Convergence of mean-field Langevin dynamics: time-space discretization, stochastic gradient, and variance reduction. Advances in Neural Information Processing Systems, 36, 15545-15577.
>
> [2] Kim, J., Yamamoto, K., Oko, K., Yang, Z., & Suzuki, T. (2023). Symmetric mean-field langevin dynamics for distributional minimax problems. arXiv preprint arXiv:2312.01127.
>
> [3] Lu, Yulong. Two-scale gradient descent ascent dynamics finds mixed Nash equilibria of continuous games: A mean-field perspective. International Conference on Machine Learning. PMLR, 2023.

---

> > ### Comment · Reviewer_eWwv · 2025-04-04
> >
> > I thank the authors for their response. Although the authors have indicated that a particle discretization analysis is forthcoming in an extended edition of this paper, I believe such a result is integral to this type of theoretical analysis. As a result, I remain lukewarm on the current draft and will opt to maintain my score.

---

> > > ### Author Response · Authors · 2025-04-08
> > >
> > > We thank the reviewer for their thoughtful feedback. We fully agree that a particle discretization analysis is important for this line of theoretical work.
> > >
> > > To draw a full picture of particle case, we write a detailed setting as follows:
> > >
> > > We set the distribution $\mu,\nu$ using $N$ finite particles $\boldsymbol \theta=[\theta^i]^N_{i=1}$ and $\boldsymbol \omega = [\omega^i]^N_{i=1}$ given by $\mu_\theta=\frac{1}{N}\sum_{i=1}^N \delta_{ \theta^i}$ and $\nu_\omega=\frac{1}{N}\sum_{i=1}^N {\delta}_{ \omega^i}$. In this case, the algorithm becomes:
> > >
> > > **For** $k=1,2,\ldots, K-1$ **do**:
> > >
> > > ​     **For** all particles $i=1,2,\ldots,N$ sample $\xi_k^{\mu,i}\sim {\cal N}(0, I_d), \xi_k^{\nu,i}\sim {\cal N}(0, I_d)$ **do**:
> > >
> > > $\theta^i_{k+1} \leftarrow \theta^i_k- \frac{\eta_1}{N} (\sum^N_{i=1} \nabla \frac{\delta J}{\delta \mu} \[\mu_k,\nu_k\](\theta^i_k)+\tau \theta^i_k)+\sqrt{2\eta_1 \tau}\xi^{\mu,i}_k.$
> > >
> > > $\omega^i_{k+1} \leftarrow \omega^i_{k} + \frac{\eta_2}{N}(\sum_{i=1}^N \nabla \frac{\delta J}{\delta \nu}\[\mu_{k+1},\nu_k\](\omega^i_k)+\tau \omega^i_k)+\sqrt{2\eta_1\tau}\xi^{\nu,i}_k.$
> > >
> > > The main differences between particle case and mean-field case focus on these four aspects:
> > >
> > > 1. Change of probability spaces: Although $\mu_\theta=\frac{1}{N}\sum_{i=1}^N \delta_{ \theta^i}$ and
> > > $\nu_{\omega}=\frac{1}{N}\sum^N_{i=1} \delta^i_\omega$ can be seen as a mixture of atom measures in space $\mathbb R^d$, however, what we focus on is the parameter space $\[ \theta^i_k \] ^N_{i=1}$ and $\[\omega^i_k \]^N_{i=1}$ is $\mathbb R^{Nd}$. So the each-step update in Lemma 1 will consider a iteration on space $\mathcal P(\mathbb R^{Nd})$, which will include subtler analysis.
> > > 2. Change of Gibbs measures: As we have said in 1, the target measure is no longer particle approximation but the parameter space $\mathcal P(\mathbb R^{Nd})$, as in Kim et. al.(2024), there will be a change in the definition of Gibbs operator,
> > > $$
> > > \mathcal K^+[\mu]\propto \mu^{\otimes N}\exp(-N\sum^N_{i=1}g_k^i).
> > > $$
> > > Hence the Log-Sobolev inequality should also change due to this issue.
> > > 4. During the update process,  the correlation between one particle and itself cannot be omitted, and the correlation will be larger if $N$ is smaller.
> > >
> > > Hence, following by our proof analysis, here are the parts that we need to modify:
> > >
> > > **Boundedness of gradients**: Since every Wasserstein gradient becomes a weighted sum of gradient of each particles, e.g. eq(38) will becomes
> > > $$
> > > \mathbb E_{\nu_k}[\\|\omega_k\\|^2_2]= \frac{1}{N}\sum^N_{i=1} \mathbb E_{\nu_k}[\\|\omega^i_k\\|^2_2]
> > > $$
> > > And the iteration will become
> > > $$
> > > \mathbb E_{\nu_k}[\\|\omega^i_k\\|^2_2]=\mathbb E_{\nu_{k-1}\otimes\rho}[\\|\omega^i_{k-1}+\eta_2h_k+\sqrt{2\eta_2\tau}\xi_k^i\\|^2_2].
> > > $$
> > > Where $h_k$ is similar in the algorithm. Hence, all the parts that contains gradient will add a term related with $N$.
> > >
> > > **Correspondence in Second-order Variation**: For example, in eq(52), when $\mu,\nu$ are all particle measures, it will become
> > > $$
> > > \frac{1}{N}\sum_{i,j=1}^N (H(\omega_{k+1}^i,\omega_{k+1}^j)-H(\omega_{k+1}^i,\omega_k^j)-H(\omega_k^i,\omega_{k+1}^i)+H(\omega_k^i,\omega_k^j)).
> > > $$
> > > When $i=j$, the dependence with particle and itself will introduce a bias term $O(\eta)$.
> > >
> > > **Failure of LSI**: Since the suboptimality of Gibbs operators, the LSI eq.(36), eq.(64) will fail.  While [1] gave a solution to this case in Lemma 7, which has form
> > > $$
> > > \frac{\tau^2}{N} \sum^N_{i=1} \mathbb E[\|h_{\omega_k}-\nabla \log \nu_{\omega_k}\|^2_2]\geq 2\alpha \tau {\mathcal L_2}(\mu_{\theta_{k+1}},\nu_{\omega_k})+O(\frac{1}{N})
> > > $$
> > > After these clarification, most of derivations we did such as Taylor expansions are safe after replacing mean-field case with particle case.  However, since all parts of our proof needs to be rewritten with form related with particle number $N$, plus the above three main differences, these additional analysis is rather involved, potentially doubling or even tripling its size. For this reason, and to maintain clarity within the scope of a conference submission, we have chosen to defer this component to an extended version. We appreciate the reviewer’s understanding and thoughtful consideration.
> > >
> > > ### References
> > > [1] Suzuki, T., Wu, D., & Nitanda, A. (2023). Convergence of mean-field Langevin dynamics: time-space discretization, stochastic gradient, and variance reduction. Advances in Neural Information Processing Systems, 36, 15545-15577.

---

### Decision · Program_Chairs · 2025-05-01

**Decision:**

Accept (spotlight poster)

**Comment:**

This paper presents a convergence analysis of the discrete-time mean-field Langevin stochastic descent-ascent (MFL-SDA) algorithm for solving distributional minimax optimization problems. The authors establish an $O(\frac{1}{\epsilon} \log(\frac{1}{\epsilon}))$ convergence rate under log-Sobolev conditions, which is nearly optimal. A key technical contribution is the direct discrete-time analysis without resorting to continuous-time PDE arguments.

The work is theoretically well-motivated and relevant to key applications, including GANs, zero-sum games, and robust learning. All reviewers leaned toward acceptance, with several increasing their scores after a detailed and thoughtful rebuttal.

That said, reviewer 5VGF pointed out a crucial issue: the dependence of convergence rate on parameters such as the dimension, regularization parameter, and LSI constant was suppressed in the initial submission. The authors responded with clear expressions, but this content should be more prominently discussed in the final version.

Overall, this paper advances the theoretical understanding of Langevin-type algorithms in minimax distributional settings and deserves to be accepted.